# QUERY-SPECIFIC GNN: A COMPREHENSIVE GRAPH REPRESENTATION LEARNING METHOD FOR RETRIEVAL AUGMENTED GENERATION

## ABSTRACT

Retrieval-augmented generation (RAG) has demonstrated its ability to enhance Large Language Models (LLMs) by integrating external knowledge sources. However, multi-hop questions, which require the identification of multiple knowledge targets to form a synthesized answer, raise new challenges for RAG systems. Under the multi-hop settings, existing methods often struggle to fully understand the questions with complex semantic structures and are susceptible to irrelevant noise during the retrieval of multiple information targets. To address these limitations, we propose a novel graph representation learning framework for multi-hop question retrieval. We first introduce a Multi-information Level Knowledge Graph (Multi-L KG) to model various information levels for a more comprehensive understanding of multi-hop questions. Based on this, we design a Query-Specific Graph Neural Network (QSGNN) for representation learning on the Multi-L KG. QSGNN employs intra/inter-level message passing mechanisms, and in each message passing the information aggregation is guided by the query, which not only facilitates multi-granular information aggregation but also significantly reduces the impact of noise. To enhance its ability to learn robust representations, we further propose two synthesized data generation strategies for pre-training the QSGNN. Extensive experimental results demonstrate the effectiveness of our framework in multi-hop scenarios, especially in high-hop questions the improvement can reach 33.8%. The code is provided by an anonymous link[1].

## 1 INTRODUCTION

Retrieval-Augmented Generation (RAG) has emerged as a powerful paradigm for enhancing the capabilities of Large Language Models (LLMs) (Peng et al., 2024; Procko & Ochoa, 2024; Gao et al., 2023). By retrieving pertinent information from external knowledge sources and integrating it into the generation process, RAG enables LLMs to ground their responses in factual information, significantly reducing the propensity for hallucinations (Fan et al., 2024).

A challenging scenario in RAG is handling multi-hop questions, where the answer cannot be directly supported by a single document[2] but requires information synthesized from multiple interrelated documents, each contributing a unique piece of the puzzle for the answer (Dua et al., 2019; Chen et al., 2019). These questions demand that the RAG system accurately identify multiple relevant documents and synthesize them to form a coherent answer. Traditional RAG approaches, which primarily rely on semantic similarity for retrieval (Karpukhin et al., 2020; Chen et al., 2024; Jiang et al., 2023; Trivedi et al., 2022b), often struggle with multi-hop questions since they overlook the relational dependencies between documents. To overcome this limitation, recent efforts have proposed to leverage Knowledge Graphs (KGs). These methods first construct KGs to model the relationships between documents. During retrieval stage, they either navigate the KGs using graph search strategies (Edge et al., 2024; Guo et al., 2024; Gutiérrez et al., 2025) or utilize Graph Neural Networks (GNNs) (Fang et al., 2019; Mavromatis & Karypis, 2024; Luo et al., 2025) to identify

---

[1] https://anonymous.4open.science/r/QSGNN-D02A

[2] Following prior works (Edge et al., 2024; Jimenez Gutierrez et al., 2024; Gutiérrez et al., 2025), we view document as the retrieval target in this paper.

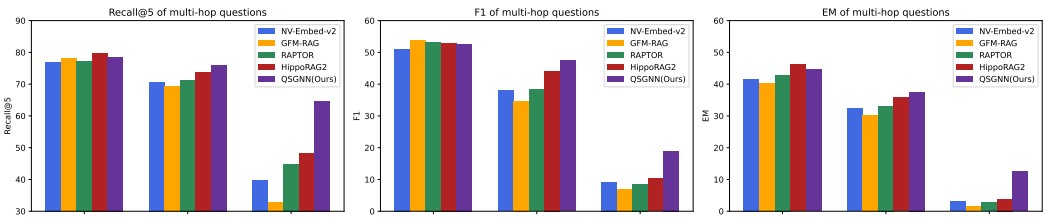

Figure 1: **Performance on multi-hop questions.** Existing methods struggle to perform well as hop number increases while QSGNN can achieve better performance on high-hop questions.

relevant information. By using KGs, these approaches can better capture the dependencies between documents, thereby improving the retrieval process for multi-hop questions.

Although KG based methods have shown improved performance over traditional approaches, they still face significant challenges in addressing the complexities of multi-hop questions. These challenges can be summarized as follows: i) *Semantic Comprehensiveness*. Multi-hop questions are inherently more complex than one-hop questions from the semantic perspective, requiring a comprehensive understanding of diverse information components. For instance, consider the question: "What was Professor Joe's opinion about the Apple event in 2023 that featured the announcement of a foldable smartphone concept?" In this case, the entity "Apple" has multiple possible meanings (*e.g.*, a company or a fruit), and the phrase "announcement of a foldable smartphone concept" may refer to a sentence or even a paragraph whose meaning cannot be fully captured by discrete entities within the KG. Such questions not only require the retrieval process to understand context-dependent entities but also demand a full understanding of multi-granular information. However, existing methods primarily focus on entities while ignoring the contextual and multi-granular information. This limitation may lead to the inclusion of irrelevant information (*e.g.*, descriptions of apples as fruits) or the loss of critical details (*e.g.*, the absence of precise entities like "foldable smartphone concept" in the KG), ultimately compromising the performance. ii) *Noise Sensitivity*. The retrieval process for multi-hop questions is highly sensitive to noise since it has multiple retrieval targets. The noise in any one retrieval target will disrupt the final result. However, existing methods are mostly prone to noise influence. For instance, graph search based methods rely on heuristic strategies to explore relevant information, which is likely to incorporate irrelevant nodes or miss important information. GNN based methods use message passing, which may also aggregate information from irrelevant neighbors. Furthermore, the large search space of KG exacerbates this issue. As shown in Figure 1, the performance of existing methods degrades significantly for high-hop questions, because the number of irrelevant nodes in the KG increases exponentially as the retrieval step increases.

To address these limitations, we propose an effective graph representation learning framework to enhance the RAG performance on multi-hop questions. Our approach is built on three key innovations: i) To comprehensively capture multiple information granularities and their complex relationships, we propose a *Multi-information Level Knowledge Graph (Multi-L KG)*. It consists of three node levels, each representing distinct information granularity. The entity-level captures the basic semantic relationships, the chunk-level models the local contextual relationships, and the document-level represents the global thematic relationships. This design encompasses the fine-grained information relationships and facilitates a more comprehensive understanding for multi-hop questions. ii) To learn representations from the Multi-L KG for multi-hop retrieval, we design a novel GNN model called *Query-Specific Graph Neural Network (QSGNN)*. Specifically, it adopts two kinds of message passing to handle the multiple information components of Multi-L KG: intra-level and inter-level message passing. Intra-level message passing focuses on the basic semantic relationships and the logical coherence relationships within each level, while inter-level message passing considers local-global relationships across levels. In each type of message passing, it considers both semantic relationships and alignment with the query. This design integrates multi-level information and significantly reduces the influence of irrelevant nodes, resulting in high-quality representations for multi-hop retrieval. iii) To further enhance the capability of QSGNN, we propose two data generation strategies to produce synthesized QA pairs for pre-training. This process requires no extra human cost and can be conducted after the construction of Multi-L KG. After pre-training, QS-GNN can be quickly adapted to downstream tasks. We conduct extensive experiments to evaluate our method, which demonstrate that *QSGNN achieves state-of-the-art performance compared to existing methods, especially in high hop questions the improvement can reach 33.8%*.

## 2 RELATED WORK

**Retrieval-augmented generation(RAG).** RAG aims at enhancing large language models (LLMs) by retrieving relevant information from external knowledge sources and integrating it into the generation process to enable fact-grounded responses (Gao et al., 2023; Fan et al., 2024). Traditional RAG methods (Chen et al., 2024; Li et al., 2023) encode documents from external knowledge sources into dense embeddings and retrieve relevant information by calculating similarity between query and document embeddings. However, these approaches fail to capture the complex relationships between documents, struggling to perform well on the multi-hop questions. In order to adapt the RAG to multi-hop questions, subsequent works (Jiang et al., 2023; Trivedi et al., 2022b; Su et al., 2024) have proposed multi-step retrieval strategies, where they decompose the original question into multiple sub-questions via LLMs and retrieve relevant information for each sub-question. Although these methods explicitly model the internal structure of the query, they still neglect the multi-level relationships between documents, resulting in limited performance in multi-hop retrieval tasks.

**Knowledge Graph for Retrieval-Augmented Generation.** Recently, Knowledge Graphs (KGs) have been proposed to facilitate multi-hop retrieval by explicitly modeling complex semantic relationships within and across documents (Peng et al., 2024; Procko & Ochoa, 2024). Current KG-based Retrieval-Augmented Generation (KG-RAG) can be categorized into two main approaches: Graph search based methods and Graph Neural Network (GNN) based methods. **Graph search based methods** identify seed nodes and leverage heuristic graph search algorithms (such as breadth-first search, community detection, or PageRank) to traverse the KG and retrieve relevant information for multi-hop questions (Edge et al., 2024; Guo et al., 2024; Jimenez Gutierrez et al., 2024; Gutiérrez et al., 2025). While these methods are intuitive, they are often susceptible to noise due to their dependence on heuristic strategies, which can introduce irrelevant nodes into the retrieval path. Additionally, the large search space of KGs amplifies the impact of noise, leading to information loss or inaccuracies. In contrast, **GNN based methods** utilize the message passing mechanism of GNNs to directly output retrieval results from the KG in an end-to-end manner (Luo et al., 2025; Wang et al., 2022) or to first identify seed nodes and then retrieve information using graph search algorithms (Mavromatis & Karypis, 2024; Fang et al., 2019; Yasunaga et al., 2021). These methods excel at modeling complex node dependencies through iterative information propagation; however, they also face challenges related to noise, as nodes may aggregate misleading information from irrelevant neighbors. Furthermore, GNN methods often struggle to capture multi-granular semantic information, limiting their effectiveness in addressing complex multi-hop questions.

## 3 APPROACH

This section presents the architecture of our proposed method, whose framework is shown in Figure 2. We first introduce the construction of the ***Multi-information Level Knowledge Graph (Multi-L KG)***. Next, we describe the detailed design of the ***Query-Specific Graph Neural Network (QSGNN)***. Finally, we present the pre-training strategy designed to enhance the performance of QSGNN in multi-hop question retrieval tasks.

### 3.1 MULTI-INFORMATION LEVEL KG CONSTRUCTION

Multi-hop questions are inherently more semantically complex than one-hop questions, as they often encompass not only simple entities but also complex expressions whose meanings may extend beyond the coverage of entities within KGs. Current approaches primarily focus on entities while neglect the multi-granular information, limiting their effectiveness in handling multi-hop questions. To address this limitation, we propose the ***Multi-information Level Knowledge Graph (Multi-L KG)***, which integrates various granularities of information to model relationships across different semantic perspectives.

We define the Multi-L KG as $G = (\mathcal{O}, \mathcal{C}, \mathcal{D}, \mathcal{E}_{oo}, \mathcal{E}_{oc}, \mathcal{E}_{od}, \mathcal{E}_{cc}, \mathcal{E}_{cd})$. It contains three types of node sets: entity set $\mathcal{O}$, chunk set $\mathcal{C}$, document set $\mathcal{D}$ and five types of edge sets: entity-entity $\mathcal{E}_{oo}$, entity-chunk $\mathcal{E}_{oc}$, entity-document $\mathcal{E}_{od}$, chunk-chunk $\mathcal{E}_{cc}$, chunk-document $\mathcal{E}_{cd}$. The construction of Multi-L KG involves two main steps:

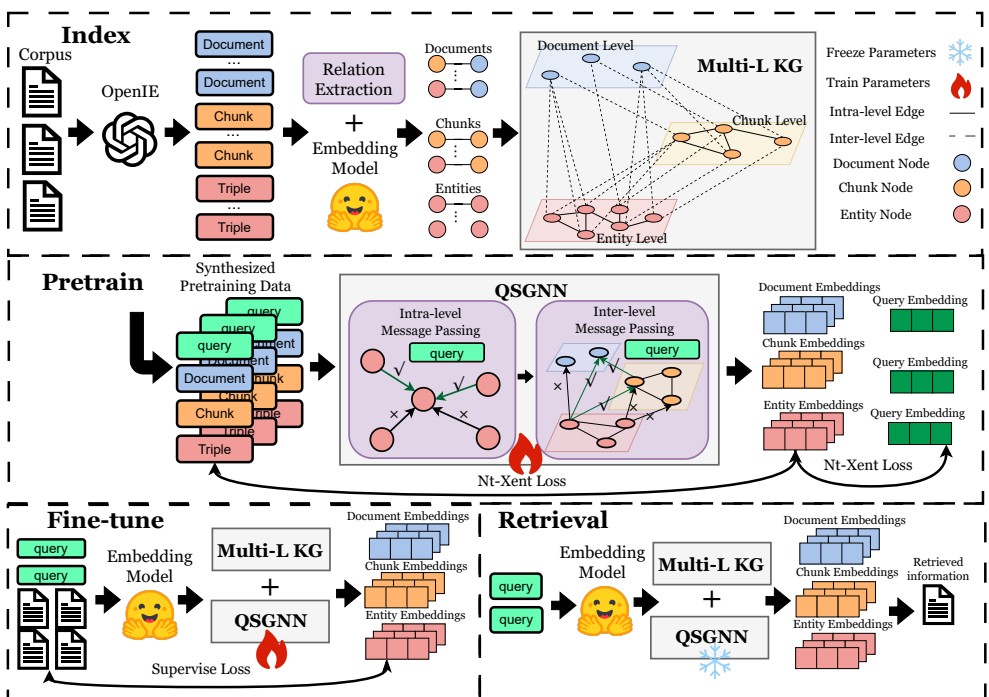

Figure 2: **Framework overview.** It first constructs Multi-L KG to model the multi-level relationships within corpora. QSGNN is designed to aggregate information from different levels, all the aggregations are guided by query. After pre-training and fine-tuning, it can generate representations for multi-hop questions.

**Node Extraction**. Given the document corpus $\mathcal{D}$, we utilize $OpenIE(.)$ (Angeli et al., 2015; Etzioni et al., 2008; Zhou et al., 2022) to extract chunks from each document and derive triples from these chunks[3]. Note that the term "chunk" can have various meanings and in this paper we view sentence as chunk because we think in most cases sentence is enough to act as a meaningful reference for questions. The subjects and the objects within the triples are collected as entities. The extraction process can be formalized as:

$$OpenIE(\mathcal{D}) \rightarrow \{(d_i, \mathcal{C}_{d_i}, \mathcal{T}_{d_i})|d_i \in \mathcal{D}\} \rightarrow \mathcal{C} = \{\mathcal{C}_{d_i}|d_i \in \mathcal{D}\}, \mathcal{O} = \{o_i|o_i \in \mathcal{T}_{d_i}, d_i \in \mathcal{D}\},$$
(1)

where $d_i$ represents the document $i$, $\mathcal{C}_{d_i}$ denotes the set of chunks extracted from $d_i$, $\mathcal{T}_{d_i}$ is the set of triples derived from $d_i$, $o_i$ refers to the subject or object within these triples $\mathcal{T}_{d_i}$.

**Relation Construction**. The entity-entity set $\mathcal{E}_{oo}$ is established based on all extracted triples $\{\mathcal{T}_{d_i}|d_i \in \mathcal{D}\}$, capturing fundamental semantic connections. The chunk-chunk set $\mathcal{E}_{cc}$ is constructed by linking adjacent chunks within a document, as their adjacency reflects a kind of logical coherent expression within a document. The last three edge sets entity-chunk $\mathcal{E}_{oc}$, entity-document $\mathcal{E}_{od}$ and chunk-document $\mathcal{E}_{cd}$ are constructed based on containment relationships.

We argue that Multi-L KG provides a comprehensive framework for modeling relationships from multiple perspectives. First, it captures both basic semantic relationships($\mathcal{E}_{oo}$) and logical coherence($\mathcal{E}_{cc}$). Second, $\mathcal{E}_{oc}$ and $\mathcal{E}_{od}$ represent local-to-global relationships, enabling precise understanding of entities in different contexts (e.g., "Apple" as a fruit or a company). Finally, $\mathcal{E}_{cd}$ serves as a high-level semantic bridge, connecting chunks to their corresponding documents for broader contextual understanding.

---

[3]The prompt for the OpenIE extraction is detailed in the Appendix A.10

## 3.2 QUERY-SPECIFIC GRAPH NEURAL NETWORK

After constructing the Multi-L KG, the next key challenge is how to effectively retrieve relevant information for multi-hop questions. Existing methods face limitations in their capacity to integrate diverse information levels, and are likely to introduce irrelevant noise due to the complexity of Multi-L KG. To address these challenges, we propose a novel ***Query-Specific Graph Neural Network (QSGNN)***. This approach adaptively aggregates information based on the query, effectively mitigating noise while incorporating insights from all information levels within the Multi-L KG, to generate comprehensive, query-specific representations for retrieval. QSGNN has two kinds of message passing process: intra-level message passing and inter-level message passing. These two processes are described in detail as follows:

**Intra-level Message Passing**. Intra-level message passing aggregates information within the same level(i.e. $\mathcal{E}_{oo}$ and $\mathcal{E}_{cc}$). For each information level, the process is defined as:

$$
\begin{aligned}
\alpha_{i,j} &= Sim\left(\mathbf{h}_i^{l-1}W_\alpha^q, \mathbf{h}_j^{l-1}W_\alpha^k\right), \\
\beta_{i,j} &= Sim\left(\mathbf{q}W_\beta^q, (\mathbf{h}_i^{l-1}||\mathbf{h}_j^{l-1})W_\beta^k\right), \\
attn &= Softmax\left(\{\alpha_{i,j}+\beta_{i,j}|j\in\mathcal{N}(i)\}\right), \\
msg_i &= \sum_{j\in\mathcal{N}(i)} attn_{i,j}\mathbf{h}_j^{l-1}W^v, \\
\mathbf{h}_i^l &= P\left(Norm\left(\mathbf{h}_i^{l-1}+msg_i\right)\right),
\end{aligned}
\tag{2}
$$

where $\mathbf{h}_{i/j}^l \in R^n$ is the representation of node $i/j$ in layer $l$, initialized with text embeddings from an embedding model [4]. $Sim(.)$ calculates the cosine similarity. $\mathbf{q} \in R^n$ is the query embedding.$||$ represents the vector concatenation. $W_{\alpha/\beta}^q \in R^{n\times n}$, $W_\alpha^k \in R^{n\times n}$, $W_\beta^k \in R^{2n\times n}$, $W^v \in R^{n\times n}$ are all learnable parameters. $Norm(.)$ is the normalization function, $P(.)$ is the 2-layers MLP with ReLU (Glorot et al., 2011).

Intra-level message passing captures two types of relationships in the Multi-L KG. One is the basic semantic relationship encoded in $\mathcal{E}_{oo}$ and the other is the logical coherence relationship represented by $\mathcal{E}_{cc}$. The aggregation considers not only connectivity, but also semantic similarity $\alpha_{i,j}$ and query alignment $\beta_{i,j}$, which minimizes noise while ensuring query-aware aggregation.

**Inter-level Message Passing**. Inter-level message passing aggregates information across different levels of Multi-L KG (i.e., $\mathcal{E}_{oc}$, $\mathcal{E}_{od}$ and $\mathcal{E}_{cd}$). For each aggregation, the process is defined as:

$$
\begin{aligned}
\mathbf{p}_{i,j} &= \left((\mathbf{h}_i^{l-1})W^t||(\mathbf{h}_j^{l-1})W^s\right), \\
\gamma_{i,j} &= Sim\left(\mathbf{q}W_\gamma^q, \mathbf{p}_{i,j}W_\gamma^k\right), \\
attn &= Softmax\left(\{\gamma_{i,j}|j\in\mathcal{N}(i)\}\right), \\
\mathbf{msg}_i &= \sum_{j\in\mathcal{N}(i)} attn_{i,j}\mathbf{h}_j^{l-1}W^v, \\
\mathbf{h}_i^l &= P\left(Norm\left(\mathbf{h}_i^{l-1}+\mathbf{msg}_i\right)\right),
\end{aligned}
\tag{3}
$$

where $W^t$ and $W^s$ are used to project heterogeneous nodes into a shared representation space. $W_\gamma^q \in R^{n\times n}$, $W_\gamma^k \in R^{2n\times n}$ are learnable parameters. The other notations follow those in Equation 2.

The inter-level message passing fuses the local, global relationships from hierarchical levels into representations, which contains comprehensive understanding towards the query.

Based on these two processes, QSGNN calculates the representations as follows:

$$
\begin{aligned}
\mathbf{H}_{intra}^l &= IntraMQ(\mathbf{q}, \mathcal{O}, \mathcal{C}, \mathcal{E}_{oo}, \mathcal{E}_{cc}, \mathbf{H}^{l-1}), \\
\mathbf{H}^l &= InterMQ(\mathbf{q}, \mathcal{O}, \mathcal{C}, \mathcal{C}, \mathcal{E}_{oc}, \mathcal{E}_{od}, \mathcal{E}_{cd}, \mathbf{H}_{intra}^l),
\end{aligned}
\tag{4}
$$

where $IntraMQ(.)$ refers to the intra-level message passing defined in Equation 2 and $InterMQ(.)$ is the inter-level message passing defined in Equation 3.

---

[4]Embeddings are compressed to dimension $n$ via linear transformations as information bottlenecks.

### 3.3 TRAINING STRATEGY FOR QSGNN

To enhance the representation learning ability of QSGNN, we first pre-train it on synthesized data and then we fine-tune it with human annotations. The pre-training data are the synthesized (question, document) pairs extracted from the same corpora used to construct the Multi-L KG. These pairs contain one-hop questions and two-hop questions, which are all generated by OpenIE [5].

**One-hop Question**. We generate one-hop questions based on the triples extracted from documents. For each triple-document pair $\{(sbj, verb, obj), d\}$, the question is generated in form of $(?, verb, obj)$ or $(sbj, verb, ?)$ where the answer is $sbj$ or $obj$ and the support document is $d$.

**Two-hop Question**. Two-hop questions are generated based on relation chains formed by shared entities across different documents. For instance, if document $d_i$ contains the triple $(sbj_i, verb_i, ent)$ and document $d_j$ contains the triple $(ent, verb_j, obj_j)$ where $ent$ is the common entity, we form the relation chain as $(sbj_i, verb_i, ent, verb_j, obj_j)$, the corresponding question is generated in the form $(?, verb_i, ent, verb_j, obj_j)$ or $(sub_i, verb_i, ent, verb_j, ?)$ where the answer is $sub_i$ or $obj_j$ and the support documents are $d_i$ and $d_j$.

The QSGNN is pre-trained on the synthesized data using the NT-Xent loss (Chen et al., 2020b):

$$\mathcal{L}_{\text{NT-Xent}} = -\frac{1}{M}\sum_{i=1}^{M}\log\frac{\exp(\text{sim}(\mathbf{q}_i, \mathbf{h}_i)/\tau)}{\sum_{j\in Neg(i)}\mathbf{1}_{[j\neq i]}\exp(\text{sim}(\mathbf{q}_i, \mathbf{h}_j)/\tau)}, \quad (5)$$

where $\mathbf{q}_i$ denotes the query embedding, $h_i$ is the representation of support document, $M$ is the batch size, $\tau$ is the temperature parameter and we set it as 1.0 in our implementation, $Neg(.)$ denotes the negative sampling and we employ a hard negative sampling strategy (Schroff et al., 2015; Xu et al., 2022) with sampling number set as 30.

After pre-training, we fine-tune QSGNN on the downstream retrieval task. In this stage, the questions are collected by humans with answers and support documents. We fine-tune QSGNN with the same loss function as Equation 5.

### 3.4 QSGNN RETRIEVAL AND GENERATION

Given a query $\mathbf{q}$, the retrieval process of QSGNN is denoted as:

$$\begin{aligned} score &= Sim(\mathbf{q}, \mathbf{H}^l), \\ \mathcal{R} &= TopK(\mathcal{D}, score), \end{aligned} \quad (6)$$

where $\mathbf{H}^l$ represents the representations calculated by Equation 4. $TopK(.)$ selects the documents with $K$-highest $score$. The retrieval results are feed into LLM as context to generate response:

$$response = LLM(query, \mathcal{R}). \quad (7)$$

## 4 EXPERIMENTAL RESULTS

### 4.1 EXPERIMENTAL SETTINGS

**Dataset**. We evaluate the effectiveness of our proposed method on three multi-hop QA benchmarks: **MuSiQue** (Trivedi et al., 2022a), **2WikiMultiHopQA(2Wiki)** (Ho et al., 2020) and **HotpotQA** (Yang et al., 2018). For a fair comparison, we use the same evaluation set(1000 samples for each dataset) as prior works (Gutiérrez et al., 2025; Jimenez Gutierrez et al., 2024; Luo et al., 2025). Additionally, we sample extra 1,000 questions from the original dataset for fine-tuning and 225 questions for testing, without overlap with the evaluation set. The basic statistics of these dataset are shown in Table 1. Details about data process are presented in Appendix A.2.

**Baselines**. We compare our method (QSGNN) with four baseline categories: **i) Naive Retriever**: we use BM25 (Robertson & Walker, 1994) and Contriever (Izacard et al., 2021) together with IR-CoT (Trivedi et al., 2022b) as the naive retrievers. **ii) Text Embedding**: we use NV-Embed-v2-7B (Lee et al., 2024) and GTE-Qwen2-7B-Instruct (Li et al., 2023) to generate embeddings for retrieval.

---

[5]The prompts can be found in Appendix A.11

Table 1: Dataset statistics.

|  | MuSiQue | 2Wiki | HotpotQA |
|---|---|---|---|
| #Entity | 118,021 | 53,153 | 86,147 |
| #Chunk | 57,887 | 23,023 | 39,830 |
| #Document | 15,803 | 7,403 | 9,811 |

**iii) Graph Search**: we use GraphRAG (Edge et al., 2024), RAPTOR (Sarthi et al., 2024), LightRAG (Guo et al., 2024) HippoRAG (Jimenez Gutierrez et al., 2024) and HippoRAG2 (Gutiérrez et al., 2025) as the graph search baselines. **iv) GNN Retriever**: we use GNN-RAG (Mavromatis & Karypis, 2024) and GFM-RAG (Luo et al., 2025) as the GNN retriever baselines. The implementation details for all baselines can be found in Appendix A.3.

**Metrics**. Follow prior works (Gutiérrez et al., 2025; Jimenez Gutierrez et al., 2024; Luo et al., 2025), we evaluate retrieval performance using recall@2 and recall@5 (top-5 retrieved documents) and Question-Answering(QA) performance with exact match (EM) and F1 scores.

**Implementation Details**. Our QSGNN is implemented with 2 layers, where each layer has one intra-leverl and one inter-level message passing process. Information dimension is set as 128. All the training uses 2 NVIDIA A100(80G) GPUs. In the pre-training stage, we use 95% samples for training and 5% samples for checkpoint selection[6]. The max epoch number is set as 5. We set learning rate as 1e-4 without weight decay. The checkpoint is saved every 2000 steps. In the fine-tuning stage, the max epoch number is set as 3. We set learning rate as 5e-4. The checkpoint is saved every 100 steps and we select checkpoint based on test set (225 samples) to evaluate the performance. We use NV-Embed-v2-7B (Lee et al., 2024) as the embedding model and we use Llama-3.3-70B-Instruct (AI@Meta., 2024) as the OpenIE model. We retrieve the top-5 documents for LLM(Llama-3.3-70B-Instruct and GPT-4o-mini (OpenAI., 2024)) as context for QA task.

## 4.2 RESULTS AND DISCUSSION

**Retrieval Task**. Table 2 summarizes the retrieval performance across baselines. We have following key observations: **i)** QSGNN outperforms all baseline methods, especially compared to other GNN based methods, it achieves +11.9%, +12.9% and +5.4% improvements on MuSiQue, 2Wiki and HotpotQA respectively and +9.8% average improvement, showing its effectiveness on multi-hop questions. **ii)** The classic retrieval algorithms (BM25/Contriever) and text embedding baselines have limited performance as they fail to capture document relationships which is crucial for answering multi-hop questions. **iii)** Although GNN based methods are designed to capture multi-hop relationships, their message passing mechanisms may inadvertently aggregate irrelevant information, which degrades retrieval effectiveness. **iv)** Graph search methods generally perform better than other baselines, but their performance is still constrained by inherent limitations. The heuristic exploration rules often overlook multi-level information and may favor specific path patterns, resulting in data bias. For instance, the performance of RAPTOR varies a lot since it leverages Gaussian Mixture Model to detect clusters in KGs, which is highly biased on data.

**QA Task**. We conduct QA experiments across the methods and the results are shown in Table 3. QSGNN outperforms other baselines due to its accurate retrieval. No Retrieval achieves low EM and F1 scores, demonstrating the importance of external knowledge for the LLM's response. Notably, LightRAG struggles to perform well on these datasets, which is consistent with other reports (Gutiérrez et al., 2025; Luo et al., 2025). The main reason is that LightRAG is incompatible with multi-hop retrieval workflows as discussed in (Luo et al., 2025). The performance of the other methods have the consistent observations with Table 2 because the retrieval accuracy directly determines the quality of response. The results of GPT-4o-mini are shown in Appendix A.4.

**Multi-hop Performance**. In order to analyze the detailed performance on multi-hop questions, we evaluate the performance of the competitive baselines on different hop numbers. The results of MuSiQue are shown in Table 4 and Figure 1. Firstly, we find that all the methods exhibit performance decline as hop number increases, because of the exponentially growing search space and noise accumulation. Secondly, GNN based methods particularly underperform in high-hop scenar-

---

[6]The checkpoint is selected based on the retrieval performance on the 5% samples using Recall@5.

Table 2: **Retrieval performance of the baselines.** Rec@2/Rec@5 denotes Recall@2/Recall@5. Average means the average performance across all the datasets. We highlight the best results with **bold** and the second best results with under line. We do not report the result of LightRAG or GraphRAG because they do not retrieve documents.

| | MuSiQue | | 2Wiki | | HotpotQA | | Average | |
|---|---|---|---|---|---|---|---|---|
| Method | Rec@2 | Rec@5 | Rec@2 | Rec@5 | Rec@2 | Rec@5 | Rec@2 | Rec@5 |
| BM25 | 32.76 | 43.62 | 55.32 | 65.31 | 57.36 | 75.82 | 48.48 | 61.58 |
| Contriever+IRCoT | 34.92 | 46.27 | 46.6 | 57.53 | 56.21 | 74.37 | 45.91 | 59.39 |
| GTE-Qwen2-7B | 51.24 | 67.72 | 66.73 | 77.78 | 77.24 | 90.15 | 65.07 | 78.55 |
| NV-Embed-v2 | 50.12 | 68.19 | 67.16 | 78.66 | **79.27** | 92.82 | 65.52 | 79.89 |
| GNN-RAG | 34.13 | 53.72 | 58.67 | 69.14 | 67.18 | 81.69 | 53.33 | 68.18 |
| GFM-RAG | 49.66 | 67.19 | 62.92 | 79.20 | 74.05 | 88.86 | 62.21 | 78.42 |
| RAPTOR | 47.68 | 69.55 | 66.45 | 82.67 | 77.83 | 93.19 | 63.99 | 81.80 |
| HippoRAG | 47.55 | 68.15 | 71.95 | 86.25 | 70.17 | 84.11 | 63.22 | 79.50 |
| HippoRAG2 | 50.42 | 72.29 | 73.31 | 88.91 | 78.25 | 92.56 | 67.33 | 84.59 |
| QSGNN(Ours) | **53.88** | **75.23** | **74.02** | **89.47** | 78.42 | **93.67** | **68.77** | **86.12** |

Table 3: **QA performance using Llama-3.3-70B-Instruct.** No Retrieval means the LLM response without external knowledge. Average means the average performance across all the datasets. We highlight the best results with **bold** and the second best results with under line.

| | MuSiQue | | 2Wiki | | HotpotQA | | Average | |
|---|---|---|---|---|---|---|---|---|
| Method | EM | F1 | EM | F1 | EM | F1 | EM | F1 |
| No Retrieval | 17.29 | 26.26 | 36.37 | 41.87 | 36.42 | 46.69 | 30.03 | 38.27 |
| BM25 | 20.38 | 28.04 | 43.15 | 49.49 | 51.26 | 61.44 | 38.26 | 46.32 |
| Contriever+IRCoT | 24.29 | 31.62 | 38.02 | 40.90 | 49.89 | 60.30 | 37.4 | 44.27 |
| GTE-Qwen2-7B | 33.60 | 40.9 | 54.11 | 59.61 | 58.61 | 70.97 | 48.77 | 57.16 |
| NV-Embed-v2 | 31.66 | 39.26 | 53.78 | 59.99 | 59.53 | 72.27 | 48.32 | 57.17 |
| GNN-RAG | 28.29 | 34.09 | 38.46 | 49.77 | 52.03 | 63.52 | 39.59 | 49.13 |
| GFM-RAG | 30.05 | 39.06 | 48.99 | 59.58 | 55.60 | 67.86 | 44.88 | 55.5 |
| RAPTOR | 32.46 | 40.30 | 48.57 | 62.15 | 60.21 | 73.25 | 47.08 | 58.57 |
| LightRAG | 6.18 | 13.66 | 6.75 | 22.50 | 12.82 | 26.25 | 8.58 | 20.80 |
| GraphRAG | 34.10 | 41.90 | 52.46 | 63.63 | 59.89 | 73.35 | 48.82 | 59.63 |
| HippoRAG2 | 35.16 | 42.44 | 56.60 | 66.18 | **60.92** | 74.06 | 50.89 | 60.89 |
| QSGNN(Ours) | **36.65** | **44.93** | **57.02** | **66.83** | 60.23 | **74.44** | **51.3** | **62.07** |

ios due to: i) the message passing aggregates the information of irrelevant nodes, ii) they fail to consider multiple information levels. Finally, as for the QSGNN, it achieves competitive performance in 2-hop questions compared to other baselines and the improvements become noticeable as the hop number increases, where it gets +33.8%, +85.8% and +231% improvements in Recall@5, F1 and EM on 4-hop questions respectively. This advantage stems from QSGNN's query-guided attention mechanism and multi-level message passing, which significantly reduce the impact of noise and fuse the multi-information levels comprehensively(see the ablation study). The result of 2Wiki can be found in Appendix A.5.

**Efficiency**. We evaluate the efficiency of retrieval across different methods, which is summarized in Table 5. Graph search based methods (HippoRAG2 and RAPTOR) exhibit higher time cost because the graph search strategies are expensive. GNN based methods (QSGNN and GFM-RAG) cost less time since message passing is an efficient way to aggregate graph information (Hamilton et al., 2017). QSGNN shows more retrieval time than GFM-RAG due to the complexity of Multi-L KG.

**Ablation Study**. To evaluate the influence of different components, we conduct ablation studies and report the results in Table 6. Our key findings are: **i)** The model's performance drops significantly

Table 4: **Performance of different hop numbers.** The best result is shown in **bold**.

| Method | MuSiQue(Recall@5) | | | MuSiQue(F1) | | | MuSiQue(EM) | | |
|---|---|---|---|---|---|---|---|---|---|
| | 2-hop | 3-hop | 4-hop | 2-hop | 3-hop | 4-hop | 2-hop | 3-hop | 4-hop |
| NV-Embed-v2 | 76.86 | 70.71 | 39.67 | 51.09 | 37.98 | 9.24 | 41.67 | 32.27 | 3.07 |
| GFM-RAG | 78.22 | 69.40 | 32.78 | 53.78 | 34.63 | 7.02 | 40.35 | 30.25 | 1.45 |
| RAPTOR | 77.39 | 71.25 | 44.86 | 53.20 | 38.35 | 8.57 | 42.82 | 33.14 | 2.76 |
| GraphRAG | - | - | - | **54.99** | 39.91 | 9.79 | 45.18 | 34.38 | 3.22 |
| HippoRAG2 | **79.89** | 73.96 | 48.32 | 53.01 | 44.17 | 10.24 | **46.2** | 35.76 | 3.78 |
| QSGNN(Ours) | 78.54 | **76.02** | **64.65** | 52.64 | **47.47** | **19.03** | 44.63 | **37.85** | **12.50** |

Table 5: **Retrieval efficiency on MuSiQue.** Time means the average retrieval time of each query.

| Method | NV-Embed-v2 | GFM-RAG | RAPTOR | HippoRAG2 | QSGNN |
|---|---|---|---|---|---|
| Recall@5 | 68.19 | 67.19 | 69.55 | 72.29 | 75.23 |
| Time(s) | 0.036 | 0.086 | 0.376 | 0.317 | 0.118 |

without entities, as they form the basic semantic relationships in Multi-L KGs. Removing entities causes substantial knowledge loss. **ii)** No intra-level message passing severely impacts performance because QSGNN loses the ability to directly capture Multi-L KG structures but have to be bridged by chunk nodes. **iii)** The absence of query alignment has a particularly negative effect on high-hop questions, where the influence of noise will be amplified as the hop number increases. **iv)** Removing chunk nodes shows relatively less impact, as document nodes can provide global contextual information to compensate for their absence. More results are shown in Appendix A.6.

Table 6: **Ablation study of QSGNN.** "w/o q_attn" means no alignment with query. "w/o intra" means no intra-level message passing. "w/o entity" and "w/o chunk" mean no entity or chunk in Multi-L KG. No inter-level message passing and No document are not reported here since we need retrieve documents.

| Method | MuSiQue(Recall@5) | | | MuSiQue(F1) | | | MuSiQue(EM) | | |
|---|---|---|---|---|---|---|---|---|---|
| | 2-hop | 3-hop | 4-hop | 2-hop | 3-hop | 4-hop | 2-hop | 3-hop | 4-hop |
| NV-Embed-v2 | 76.86 | 70.71 | 39.67 | 51.09 | 37.98 | 9.24 | 41.67 | 32.27 | 3.07 |
| HippoRAG2 | 79.89 | 73.96 | 48.32 | 53.01 | 44.17 | 10.24 | 46.20 | 35.76 | 3.78 |
| QSGNN | 78.54 | 76.02 | 64.65 | 52.64 | 47.47 | 19.03 | 44.63 | 37.85 | 12.50 |
| QSGNN(w/o entity) | 60.34 | 43.31 | 25.02 | 27.25 | 16.56 | 2.05 | 19.58 | 9.65 | 1.13 |
| QSGNN(w/o chunk) | 77.65 | 74.99 | 62.60 | 51.18 | 45.82 | 16.55 | 42.74 | 35.69 | 9.08 |
| QSGNN(w/o q_attn) | 76.26 | 71.51 | 53.74 | 48.02 | 40.54 | 9.65 | 40.24 | 30.29 | 6.56 |
| QSGNN(w/o intra) | 71.33 | 68.42 | 43.15 | 39.88 | 33.39 | 6.93 | 28.76 | 22.85 | 2.61 |

**Other Discussion**. For the page limitation, we discuss the hyperparameter influence of QSGNN in Appendix A.7. We conduct the good case study in Appendix A.8. The bad case study and the discussion of limitations for QSGNN can be found in Appendix A.9.

## 5 CONCLUSION

In this paper, we propose a novel graph representation learning framework for multi-hop questions retrieval. We first design a Multi-information Level Knowledge Graph (Multi-L KG) to model the multi-granular relationships among documents. Then we introduce Query-Specific Graph Neural Network (QSGNN). The QSGNN aggregates information from multi-levels, together with query alignment, to ensure comprehensive query-specific representations while filtering out the noise. The pre-training strategy further enhance the ability of QSGNN. Extensive experiments shows the effectiveness of our method in multi-hop questions.

**Ethics Statement**. We affirm that our research adheres to the highest ethical standards in the conduct of scientific research. This work does not involve human subjects, animal testing, harmful insights, discrimination, sensitive personal data, or any legal compliance problem. All experiments were conducted using publicly available datasets or Large Language Model (LLM) tools, and we have taken steps to ensure the reproducibility and transparency of our results. We have considered potential societal impacts of our research and believe that the proposed methods and findings contribute positively to the field of LLMs and its applications.

**Reproducibility Statement**. To ensure the reproducibility of our work, we provide the following statement. All code implementations of our proposed method are provided by an anonymous link, and we will make them publicly open source upon the acceptance of this paper. Additionally, all the large language models (LLMs) used in this paper are open source, as detailed in prior work (Gutiérrez et al., 2025; Jimenez Gutierrez et al., 2024; Luo et al., 2025). Regarding the data used in our experiments, we follow the standard protocols established by open source works (Gutiérrez et al., 2025). All the data processing details are presented in Appendix A.2. For better reproducibility, we also include example data in our code repository, which will be released publicly after the acceptance of the paper.

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

# A APPENDIX

This supplementary material provides additional details on the proposed method and experimental results that could not be included in the main manuscript due to page limitations. Specifically, this appendix is organized as follows.

- Sec. A.1 provides the use of Large Language Models (LLMs).
- Sec. A.2 provides more details on the dataset settings and data processing.
- Sec. A.3 outlines the implementation of baseline models and their experimental settings.
- Sec. A.4 presents the QA performance on GPT-4o-mini.
- Sec. A.5 provides the multi-hop performance on 2WikimultihopQA(2Wiki) dataset.
- Sec. A.6 provides detailed ablation study for our method.
- Sec. A.7 presents hyperparameter study for our method.
- Sec. A.8 discusses three good cases of our method compared to HippoRAG2
- Sec. A.9 presents three bad cases of our method and discusses the limitation and possible solution as future work.
- Sec. A.10 presents the prompts used for OpenIE.
- Sec. A.11 presents the prompts used for generating pre-training data.

## A.1 THE USE OF LARGE LANGUAGE MODELS

We present the statement regarding the usage of Large Language Models (LLMs) in our paper. We assert that **there is no involvement of LLMs in the innovation, implementation, or paper writing of our method**. However, we still use LLMs in two ways:

i) Our proposed method is about the Retrieval-augmented generation (RAG) for LLMs. Therefore, in this paper, we use LLMs as the QA performer and the OpenIE tool to evaluate the effectiveness of our method, similar to other LLM-related works (Guo et al., 2024; Jimenez Gutierrez et al., 2024; Luo et al., 2025).

ii) We use LLMs to help us correct some grammatical errors in our paper writing.

## A.2 DATASET SETTINGS

Our dataset is constructed based on the benchmark datasets used by prior works (Jimenez Gutierrez et al., 2024; Gutiérrez et al., 2025; Luo et al., 2025). Here we present the construction details.

**Dataset Construction**. The evaluation set is the same as HippoRAG2(Gutiérrez et al., 2025), which is the state-of-the-art baseline. And the corpora is expanded. Since our method need extra QA pairs for fine-tuning, we randomly sample extra samples from each original dataset and we make sure there is no overlap with the evaluation set. For the MuSiQue, the sample should have at least 60% supported documents contained in existing corpora. For the 2Wiki, the sample should have at least 60% overlap with the corpora. For the HotpotQA, the overlap ratio is 50%. Within these extra samples, we randomly select 1000 questions for fine-tuning and 225 questions for testing. The corpora of each dataset will be expanded to cover the new samples related documents(supported documents and distracted documents).

**Pre-training Data Construction**. We generate synthesized pre-training data for each dataset. For the MuSiQue the one-hop QA pairs are 91,621 and the two-hop QA pairs are 58,923. For the 2Wiki, the one-hop QA pairs are 52,122 and the two-hop QA pairs are 12,097. For the HotpotQA, the one-hop QA pairs are 73,128 and the two-hop QA pairs are 20,619. We combine and shuffle all these QA pairs of each dataset as pre-training data.

The statistics of these datasets are shown in Table 7.

Table 7: Dataset statistics

|  | MuSiQue | 2Wiki | HotpotQA |
|---|---|---|---|
| #Entity | 118,021 | 53,153 | 86,147 |
| #Chunk | 57,887 | 23,023 | 39,830 |
| #Document | 15,803 | 7,403 | 9,811 |
| #Pre-train 1-hop QA | 91,621 | 52,122 | 73,128 |
| #Pre-train 2-hop QA | 58,923 | 12,097 | 20,619 |
| #Pre-train QA | 150,544 | 64,219 | 93,747 |
| #Fine-tune 2-hop QA | 270 | 782 | 1000 |
| #Fine-tune 3-hop QA | 482 | 0 | 0 |
| #Fine-tune 4-hop QA | 248 | 218 | 0 |
| #Fine-tune QA | 1,000 | 1000 | 1000 |
| #Test 2-hop QA | 66 | 172 | 225 |
| #Test 3-hop QA | 104 | 0 | 0 |
| #Test 4-hop QA | 55 | 53 | 0 |
| #Test QA | 225 | 225 | 225 |
| #Eval 2-hop QA | 488 | 847 | 1000 |
| #Eval 3-hop QA | 334 | 0 | 0 |
| #Eval 4-hop QA | 178 | 153 | 0 |
| #Eval QA | 1,000 | 1000 | 1000 |

### A.3 BASELINE SETTINGS

**Implementation**. The BM25 retrieval algorithm is implemented as BM25S (Lù, 2024). The Contriever (Izacard et al., 2021) is implemented with the official code acting as the retrieval server in IRCoT (Trivedi et al., 2022b). As for the text embedding based model, we use the NV-Embed-v2-7B and GTE-Qwen2-7B-Instruct from Huggingface (Wolf et al., 2019). For the all the GNN based methods and graph search based methods we use their official implementations.

**Experimental Settings**. For all the baselines, we conduct experiment under the official settings or prior works experimental suggestions. Specifically, their settings are described as follows:

**i)** For BM25, we retrieve 5 most relevant documents based on the query terms frequency appearing in each document.

**ii)** For Contriever+IRCoT, we follow the suggestion of HippoRAG2 (Gutiérrez et al., 2025) that we retrieve the top 10 documents for each step, the maximum step number for MuSiQue is set as 4 and the maximum step number for other datasets is set as 2.

**iii)** For text embedding based methods, embeddings are calculated for all the documents within our datasets, we retrieve the 5 most relevant documents as (Jimenez Gutierrez et al., 2024).

**iv)** For GFM-RAG, we use the official model implementation where we pre-train and fine-tune the GNN on our corpora and all the hyperparameters are set as (Luo et al., 2025) suggests. We retrieve the 5 most relevant documents.

**v)** For GNN-RAG, we use ReaRev (Mavromatis & Karypis, 2022) as GNN reasoning, it samples dense subgraph by PageRank (Andersen et al., 2006) on our datasets, all the settings are the same as the official settings (Mavromatis & Karypis, 2024).

**vi)** For GraphRAG and LightRAG, the implementations are based on the official codes, and the hyperparameters are set as HippoRAG2 (Gutiérrez et al., 2025) suggests.

**vii)** For RAPTOR, HippoRAG and HippoRAG2, the settings are set the same as their official implementations.

For all the methods, we use LLM (Llama-3.3-70B-Instruct (AI@Meta., 2024) and GPT-4o-mini (OpenAI., 2024)) for QA task.

## A.4 QA Performance on GPT-4o-mini

Table 8: QA performance on GPT-4o-mini. Average means the average performance across all the datasets. We highlight the best results with **bold** and the second best results with under line.

| Method | MuSiQue | | 2Wiki | | HotpotQA | | Average | |
|---|---|---|---|---|---|---|---|---|
| | EM | F1 | EM | F1 | EM | F1 | EM | F1 |
| No Retrieval | 10.81 | 21.39 | 31.83 | 36.51 | 27.44 | 41.47 | 23.36 | 33.12 |
| BM25 | 12.54 | 23.88 | 32.25 | 40.30 | 34.50 | 52.62 | 26.43 | 38.93 |
| Contriever+IRCoT | 13.90 | 25.28 | 29.44 | 34.66 | 30.17 | 45.55 | 24.50 | 35.16 |
| GTE-Qwen2-7B | 31.15 | 39.88 | 50.75 | 52.37 | 52.19 | 66.17 | 44.70 | 52.81 |
| NV-Embed-v2 | 28.07 | 37.15 | 49.22 | 51.51 | 54.85 | 68.43 | 44.05 | 52.36 |
| GNN-RAG | 23.02 | 30.12 | 34.52 | 48.25 | 49.12 | 58.99 | 35.55 | 45.79 |
| GFM-RAG | 28.30 | 36.81 | 43.79 | 53.26 | 51.86 | 64.73 | 41.32 | 51.6 |
| RAPTOR | 32.93 | 41.04 | 45.43 | 56.24 | 55.19 | 68.16 | 44.52 | 55.15 |
| GraphRAG | 33.31 | 40.13 | 50.99 | 58.08 | 56.45 | 69.24 | 46.92 | 55.82 |
| HippoRAG2 | 33.12 | 40.30 | 51.49 | 60.18 | **56.39** | **70.53** | 47.00 | 57.00 |
| QSGNN(Ours) | **34.27** | **41.78** | **53.52** | **61.63** | 56.04 | 69.73 | **47.94** | **57.71** |

**QA Task on GPT-4o-mini**. Table 8 presents QA performance using GPT-4o-mini. We do not report the performance of LightRAG as it is unsuitable for multi-hop QA tasks. The results show that QS-GNN achieves competitive EM and F1 scores, demonstrating its compatibility with different LLM backbones. No Retrieval gets the lowest EM and F1 score due to the limited knowledge of GPT-4o-mini. Notably, all methods show reduced performance compared to Llama-3-70B-Instruct(see Table 3), primarily due to the intrinsic knowledge gap between GPT-4o-mini and Llama-3.3-70B-Instruct. Other observations align with the report of Table 3.

## A.5 Multi-hop Performance on 2Wiki

Table 9: Performance of different hop numbers on 2Wiki. The best result is shown in **bold**.

| Method | 2Wiki(Recall@5) | | 2Wiki(F1) | | 2Wiki(EM) | |
|---|---|---|---|---|---|---|
| | 2-hop | 4-hop | 2-hop | 4-hop | 2-hop | 4-hop |
| NV-Embed-v2 | 81.40 | 63.52 | 63.21 | 42.19 | 59.28 | 23.35 |
| GFM-RAG | 83.37 | 56.09 | 64.36 | 33.11 | 54.44 | 18.82 |
| RAPTOR | 85.91 | 64.75 | 66.13 | 40.14 | 53.71 | 20.12 |
| GraphRAG | - | - | 67.99 | 39.52 | 57.24 | 25.97 |
| HippoRAG2 | **93.15** | 65.44 | **70.50** | 42.26 | **61.53** | 29.32 |
| QSGNN | 92.10 | **74.88** | 69.72 | **50.82** | 60.04 | **40.28** |

The multi-hop performance on 2Wiki dataset is presented in Table 9. We find that QSGNN can achieve +14.4%, +20.2% and +37.3% on Recall@5, F1 and EM respectively. For the other baselines, we have the same observations as Table 4.

## A.6 Ablation Study

We present the ablation study of QSGNN on Table 10. As for the chunk retrieval (QSGNN + chunk, w/o inter or w/o doc), we retrieve the top 10 most relevant chunks for QA and we only report the EM and F1 score since we have no ground truth for chunk retrieval. We find that QSGNN(chunk) does not perform as well as QSGNN. It can be attributed to two factors: i) QSGNN is not directly trained on chunk labels. ii) Document generally contains more information than chunk. No inter-level message passing will not update the representations thus have no difference with the embeddings baselines, whose performance decrease a lot when hop number increases. No document lacks global

Table 10: Ablation study of QSGNN. "chunk" means performing QA task with chunk. "w/o inter" means no inter-level message passing. "w/o doc" means no document in Multi-L KG. "w/o pt" means no pre-training. "w/o ft" means no fine-tuning.

| Method | MuSiQue(Recall@5) | | | MuSiQue(F1) | | | MuSiQue(EM) | | |
|---|---|---|---|---|---|---|---|---|---|
| | 2-hop | 3-hop | 4-hop | 2-hop | 3-hop | 4-hop | 2-hop | 3-hop | 4-hop |
| NV-Embed-v2 | 76.86 | 70.71 | 39.67 | 51.09 | 37.98 | 9.24 | 41.67 | 32.27 | 3.07 |
| HippoRAG2 | 79.89 | 73.96 | 48.32 | 53.01 | 44.17 | 10.24 | 46.20 | 35.76 | 3.78 |
| QSGNN | 78.54 | 76.02 | 64.65 | 52.64 | 47.47 | 19.03 | 44.63 | 37.85 | 12.50 |
| QSGNN(w/o entity) | 60.34 | 43.31 | 25.02 | 27.25 | 16.56 | 2.05 | 19.58 | 9.65 | 1.13 |
| QSGNN(w/o chunk) | 77.65 | 74.99 | 62.60 | 51.18 | 45.82 | 16.55 | 42.74 | 35.69 | 9.08 |
| QSGNN(w/o q_attn) | 76.26 | 71.51 | 53.74 | 48.02 | 40.54 | 9.65 | 40.24 | 30.29 | 6.56 |
| QSGNN(w/o intra) | 71.33 | 68.42 | 43.15 | 39.88 | 33.39 | 6.93 | 28.76 | 22.85 | 2.61 |
| QSGNN(chunk) | - | - | - | 40.18 | 37.55 | 13.26 | 30.25 | 29.41 | 8.71 |
| QSGNN(w/o inter) | - | - | - | 40.27 | 35.98 | 4.24 | 28.36 | 19.19 | 2.72 |
| QSGNN(w/o doc) | - | - | - | 36.54 | 32.49 | 8.01 | 26.19 | 26.57 | 5.02 |
| QSGNN(w/o pt) | 63.66 | 53.93 | 36.83 | 35.64 | 28.14 | 5.15 | 24.62 | 19.28 | 2.53 |
| QSGNN(w/o ft) | 77.19 | 74.69 | 62.57 | 51.08 | 46.42 | 17.12 | 43.58 | 34.23 | 10.48 |

context information, so it performs worse than QSGNN(chunk). In addition, QSGNN(w/o ft) can achieve competitive performance in 2-hop questions but fail to perform as well as QSGNN, since we pre-train QSGNN only on 1,2-hop questions. QSGNN(w/o pt) struggles to perform well because it can not be fitted well on only small number of training pairs.

## A.7 HYPERPARAMETER DISCUSSION

Here we discussion some of important hyperparameters, including the receptive field (QSGNN layer number), negative sampling number, pre-training data scale and information bottleneck dimension.

**Receptive Field Discussion**. We discuss the relationships between GNN layers and the multi-hop performance, the results are shown in Table 11. We find that 1-layer QSGNN (one intra-level + one inter-level) have limited performance and the gap between 2-layer QSGNN becomes bigger as the hop number increase, it is because 1-layer QSGNN can only aggregate 2-hop information. The receptive field limits the potential of QSGNN. The 2-layer QSGNN achieves the best performance among 2,3,4 hop questions because it can aggregate more useful information. However, as the layer number increases(4 layers), the performance will suffer a lot because of the over-smoothing problem of GNN (Rusch et al., 2023; Chen et al., 2020a).

Table 11: The influence of GNN layer number.

| Method | MuSiQue(Recall@5) | | | MuSiQue(F1) | | | MuSiQue(EM) | | |
|---|---|---|---|---|---|---|---|---|---|
| | 2-hop | 3-hop | 4-hop | 2-hop | 3-hop | 4-hop | 2-hop | 3-hop | 4-hop |
| HippoRAG2 | 79.89 | 73.96 | 48.32 | 53.01 | 44.17 | 10.24 | 46.20 | 35.76 | 3.78 |
| QSGNN(1 layer) | 73.81 | 69.14 | 55.13 | 46.25 | 41.86 | 11.46 | 38.29 | 29.60 | 7.83 |
| QSGNN(2 layers) | 78.54 | 76.02 | 64.65 | 52.64 | 47.47 | 19.03 | 44.63 | 37.85 | 12.50 |
| QSGNN(3 layers) | 76.35 | 74.55 | 61.66 | 51.34 | 45.24 | 16.45 | 42.11 | 34.50 | 9.13 |
| QSGNN(4 layers) | 62.60 | 58.29 | 35.37 | 38.93 | 29.15 | 4.61 | 26.68 | 18.68 | 2.03 |

**Negative Sampling Number**. Table 12 shows the impact of negative sampling number on model's performance. According to the results, we find that low sampling number (Neg_N = 1) has limited performance due to inadequate discrimination between target documents and distractions. High sampling number (Neg_N=50, 100) shows limited benefits, because excessive easy negatives fail to provide meaningful guidance for QSGNN training.

Table 12: The influence of negative sampling number.

| Method | MuSiQue(Recall@5) | | | MuSiQue(F1) | | | MuSiQue(EM) | | |
|---|---|---|---|---|---|---|---|---|---|
| | 2-hop | 3-hop | 4-hop | 2-hop | 3-hop | 4-hop | 2-hop | 3-hop | 4-hop |
| HippoRAG2 | 79.89 | 73.96 | 48.32 | 53.01 | 44.17 | 10.24 | 46.20 | 35.76 | 3.78 |
| QSGNN(Neg_N=1) | 68.67 | 60.27 | 50.82 | 43.72 | 38.25 | 5.18 | 34.87 | 27.44 | 2.10 |
| QSGNN(Neg_N=10) | 77.97 | 76.16 | 62.14 | 51.03 | 46.68 | 17.62 | 42.28 | 38.54 | 10.76 |
| QSGNN(Neg_N=30) | 78.54 | 76.02 | 64.65 | 52.64 | 47.47 | 19.03 | 44.63 | 37.85 | 12.50 |
| QSGNN(Neg_N=50) | 77.18 | 76.69 | 63.51 | 51.34 | 46.99 | 18.36 | 44.07 | 38.02 | 11.65 |
| QSGNN(Neg_N=100) | 77.09 | 76.16 | 63.85 | 52.95 | 46.37 | 18.22 | 43.16 | 37.52 | 11.53 |

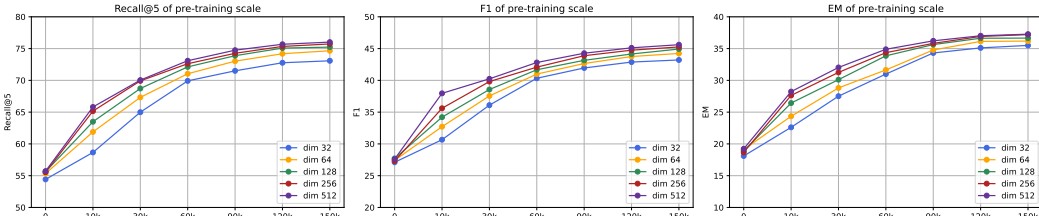

Figure 3: **Influence of pre-training scale on various model dimension.**

**Pre-training Scale and Information Dimension**. We pre-train QSGNN on the sythesized QA pairs from 0 to 150k across information dimension from 32 to 512 respectively, the results are shown in Figure 3. We conduct experiments on the MuSiQue dataset. As for the pre-training scale, the results show that insufficient pre-training data leads to sub-optimal performance across all the dimensions. As the amount of pre-training data increases, the performance of QSGNN gets better, although the marginal improvement decreases. As for the information dimension, the results indicate that increasing the information dimension improves the performance of QSGNN, as larger dimensions enhance model expressiveness through increased model scale. This observation is consistent with the model scaling analysis presented in GFM-RAG (Luo et al., 2025). We believe that further improvements can be achieved by combining a larger model with a substantial amount of pre-training data. In this paper, we set the dimension as 128 considering the balance between performance and computational limitation.

### A.8 GOOD CASE STUDIES

We present three typical comparative cases between QSGNN and HippoRAG2 to demonstrate QS-GNN's advantages. In each case, correct retrieval evidence is marked in green and incorrect evidence in red. Notably, QSGNN's marks are based on query alignment, whereas HippoRAG2's marks is based on PPR scores (Gutiérrez et al., 2025).

**Good Case 1**. Figure 4 illustrates a scenario where QSGNN achieves recall@5 of 1.0 with correct answer generation, while HippoRAG2 fails. The 4-hop question requires understanding composite semantics: **i)** "the only group larger than" corresponds to "the world's second largest recorded music company", **ii)** "the headquarters of" refers to "based in Santa Monica, California" and **iii)** "the explorer reach the city" denotes "set foot in the area was the party of explorer Gaspar de Portola". QSGNN successfully identifies the critical documents describing Sony Music Entertainment's ranking, location and the city information, whereas HippoRAG2 erroneously retrieves information about Jive Records' headquarters. This failure stems from HippoRAG2's selection of seed nodes ("headquarter", "located in") without high-level contextual information. The case validates Multi-L KG's semantic modeling capability and query alignment's effectiveness in filtering misinformation.

**Good Case 2**. Figure 5 shows QSGNN achieving perfect recall@5 while retrieval of HippoRAG2 is inaccurate. For the 4-hop question about Andrew Deveaux's birthplace, QSGNN integrates evidence across multiple documents and each document describes one key evidence for the question: **i)** biographical data ("born in South Carolina"), **ii)** state capital history ("Columbia became capital

**Query:** When did the explorer reach the city where the headquarters of the only group larger than Vilaiyaadu Mankatha's record label is located?
**Answer:** August 3, 1769

**QSGNN**
**Retrieval:**
Vilaiyaadu Mankatha\nFour songs were included as bonus tracks to the single release of \"Vilaiyaadu Mankatha\", all of which were part of earlier soundtracks by Yuvan Shankar Raja and were marketed by Sony Music Entertainment. The four songs - \"Dia Dia Dole\" performed by Suchitra for the film \"Avan Ivan\", \"Goa\" from the same-titled film featuring vocals by Krish, Ranjith, Tanvi Shah, Suchitra, Chynk Showtyme and Pav Bundy, \"Yogi Yogi Thaan\" from \"Yogi\" sung by Blaaze and Neha Bhasin and the title track from \"Theeradha Vilaiyattu Pillai\" rendered by Andrea Jeremiah, Tanvi Shah, Vinaitha and Ranjith - were added in their original composition without any variation.

Sony Music\nSony Music Entertainment Inc. (sometimes known as Sony Music or by the initials, SME) is an American music corporation managed and operated by Sony Corporation of America (SCA), a subsidiary of Japanese conglomerate Sony Corporation. In 1929, the enterprise was first founded as American Record Corporation (ARC) and, in 1938, was renamed Columbia Recording Corporation, following ARC's acquisition by CBS. In 1966, the company was reorganized to become CBS Records. In 1987, Sony Corporation of Japan bought the company, and in 1991, renamed it SME. It is the world's second largest recorded music company, after Universal Music Group.

The Right Stuff Records\nThe Right Stuff Records is a reissue record label that was part of EMI, which is now owned by Universal Music Group and is based out of Santa Monica, California.

Santa Monica, California\nSanta Monica was long inhabited by the Tongva people. Santa Monica was called Kecheek in the Tongva language. The first non-indigenous group to set foot in the area was the party of explorer Gaspar de Portolà, who camped near the present day intersection of Barrington and Ohio Avenues on August 3, 1769. There are two different versions of the naming of the city. One says that it was named in honor of the feast day of Saint Monica (mother of Saint Augustine), but her feast day is actually May 4. Another version says that it was named by Juan Crespí on account of a pair of springs, the Kuruvungna Springs (Serra Springs), that were reminiscent of the tears that Saint Monica shed over her son's early impiety.

**Answer:** August 3, 1769

**HippoRAG2**
**Retrieval:**
Vilaiyaadu Mankatha\nFour songs were included as bonus tracks to the single release of \"Vilaiyaadu Mankatha\", all of which were part of earlier soundtracks by Yuvan Shankar Raja and were marketed by Sony Music Entertainment. The four songs - \"Dia Dia Dole\" performed by Suchitra for the film \"Avan Ivan\", \"Goa\" from the same-titled film featuring vocals by Krish, Ranjith, Tanvi Shah, Suchitra, Chynk Showtyme and Pav Bundy, \"Yogi Yogi Thaan\" from \"Yogi\" sung by Blaaze and Neha Bhasin and the title track from \"Theeradha Vilaiyattu Pillai\" rendered by Andrea Jeremiah, Tanvi Shah, Vinaitha and Ranjith - were added in their original composition without any variation.

Kathmandu\nKathmandu(/ˌkɑːtmɑːnˈduː/; Nepali pronunciation: [kɑʈʰmɑɳɖu]) is the capital and largest municipality of Nepal. It also hosts the headquarters of the South Asian Association for Regional Cooperation (SAARC). It is the only city of Nepal with the administrative status of Mahanagar (Metropolitan City), as compared to Upa-Mahanagar (Sub-Metropolitan City) or Nagar (City). Kathmandu is the core of Nepal's largest urban agglomeration located in the Kathmandu Valley consisting of Lalitpur, Kirtipur, Madhyapur Thimi, Bhaktapur and a number of smaller communities. Kathmandu is also known informally as \"KTM\" or the \"tri-city\". According to the 2011 census, Kathmandu Metropolitan City has a population of 975,453 and measures 49.45 km2 (19.09 sq mi).

Francisco de Orellana\nFrancisco de Orellana (; 1511 – November 1546) was a Spanish explorer and conquistador. He completed the first known navigation of the entire length of the Amazon River, which initially was named \"Rio de Orellana.\" He also founded the city of Guayaquil in what is now Ecuador.

Jive Records\nJive Records was an American record label under the RCA Music Group formed in 1981 by Zomba Records. Formerly headquartered in New York City, the label was best known for a string of successes with hip hop artists in the 1980s, and also in teen pop and boy bands during the 1990s and early 2000s.

**Answer:** 24 Augest, 1542

Figure 4: **Good case 1 for QSGNN.**

**Query:** What county is the city that shares a border with the state capital of the state where Andrew Deveaux was born located in?
**Answer:** Richland County

**QSGNN**
**Retrieval:**
Andrew Deveaux\nAndrew Deveaux (30 April 1758 – 11 July 1812) was an American Loyalist from South Carolina who is most famous for his recapture of the Bahamas in 1783.

Charleston, South Carolina\nAlthough the city lost the status of state capital to Columbia in 1786, Charleston became even more prosperous in the plantation-dominated economy of the post-Revolutionary years. The invention of the cotton gin in 1793 revolutionized the processing of this crop, making short-staple cotton profitable. It was more easily grown in the upland areas, and cotton quickly became South Carolina's major export commodity. The Piedmont region was developed into cotton plantations, to which the sea islands and Lowcountry were already devoted. Slaves were also the primary labor force within the city, working as domestics, artisans, market workers, and laborers.

WWNQ\nWWNQ is a radio station licensed to Forest Acres, South Carolina, serving the Columbia, South Carolina market. Owned by Midlands Media Group LLC, the station broadcasts a country music format branded as 94.3 The Dude.

Forest Acres, South Carolina\nForest Acres is a city in Richland County, South Carolina, United States. The population was 10,361 at the 2010 census. It is part of the Columbia, South Carolina, Metropolitan Statistical Area.

**Answer:** Richland County

**HippoRAG2**
**Retrieval:**
Andrew Deveaux\nAndrew Deveaux (30 April 1758 – 11 July 1812) was an American Loyalist from South Carolina who is most famous for his recapture of the Bahamas in 1783.

Columbia, South Carolina\nColumbia is the capital and second largest city of the U.S. state of South Carolina, with a population estimate of 134,309 as of 2016. The city serves as the county seat of Richland County, and a portion of the city extends into neighboring Lexington County. It is the center of the Columbia metropolitan statistical area, which had a population of 767,598 as of the 2010 United States Census, growing to 817,488 by July 1, 2016, according to 2015 U.S. Census estimates. The name Columbia is a poetic term used for the United States, originating from the name of Christopher Columbus.",

Savannah, Georgia\nSavannah (/səˈvænə /) is the oldest city in the U.S. state of Georgia and is the county seat of Chatham County. Established in 1733 on the Savannah River, the city of Savannah became the British colonial capital of the Province of Georgia and later the first state capital of Georgia. A strategic port city in the American Revolution and during the American Civil War, Savannah is today an industrial center and an important Atlantic seaport. It is Georgia's fifth - largest city and third - largest metropolitan area.",

Edisto Beach State Park\nEdisto Beach State Park is located on the coast of South Carolina, 50 miles south of Charleston, near the town of Edisto Beach in Colleton County.

**Answer:** Richland County

Figure 5: **Good case 2 for QSGNN.**

**Query:** How long had the city where the Yongle Emperor greeted the Karmapa been the capitol city of Yaxing Coach's headquarters location?
**Answer:** about 400 years

**QSGNN:**
**Retrieval:**
Sino-Tibetan relations during the Ming dynasty\nDuring his travels beginning in 1403, Deshin Shekpa was induced by further exhortations by the Ming court to visit Nanjing by April 10, 1407. Norbu writes that the Yongle Emperor, following the tradition of Mongol emperors and their reverence for the Sakya lamas, showed an enormous amount of deference towards Deshin Shekpa. The Yongle Emperor came out of the palace in Nanjing to greet the Karmapa and did not require him to kowtow like a tributary vassal. According to Karma Thinley, the emperor gave the Karmapa the place of honor at his left, and on a higher throne than his own. Rossabi and others describe a similar arrangement made by Kublai Khan and the Sakya Phagpa lama, writing that Kublai would \"sit on a lower platform than the Tibetan cleric\" when receiving religious instructions from him.

Nanjing\nArchaeological discovery shows that \"Nanjing Man\" lived in more than 500 thousand years ago. Zun, a kind of wine vessel, was found to exist in Beiyinyangying culture of Nanjing in about 5000 years ago. In the late period of Shang dynasty, Taibo of Zhou came to Jiangnan and established Wu state, and the first stop is in Nanjing area according to some historians based on discoveries in Taowu and Hushu culture. According to legend,[which?] Fuchai, King of the State of Wu, founded a fort named Yecheng (冶城) in today's Nanjing area in 495 BC. Later in 473 BC, the State of Yue conquered Wu and constructed the fort of Yuecheng (越城) on the outskirts of the present-day Zhonghua Gate. In 333 BC, after eliminating the State of Yue, the State of Chu built Jinling Yi (金陵邑) in the western part of present-day Nanjing. It was renamed Moling (秣陵) during reign of Qin Shi Huang. Since then, the city experienced destruction and renewal many times.[citation needed] The area was successively part of Kuaiji, Zhang and Danyang prefectures in Qin and Han dynasty, and part of Yangzhou region which was established as the nation's 13 supervisory and administrative regions in the 5th year of Yuanfeng in Han dynasty (106 BC). Nanjing was later the capital city of Danyang Prefecture, and had been the capital city of Yangzhou for about 400 years from late Han to early Tang.

Yaxing Coach\nYaxing Coach (Yangzhou Yaxing Motor Coach Co., Ltd) is a bus manufacturer based in Yangzhou, Jiangsu, China. It is a subsidiary of Jiangsu Yaxing that was founded in 1998. Buses are produced under the \"Yaxing\", \"Yangtse (Yangzlv) \", and more recently Asiastar brands.

**Answer:** about 400 years

**HippoRAG2**
**Retrieval:**
Sino-Tibetan relations during the Ming dynasty\nDuring his travels beginning in 1403, Deshin Shekpa was induced by further exhortations by the Ming court to visit Nanjing by April 10, 1407. Norbu writes that the Yongle Emperor, following the tradition of Mongol emperors and their reverence for the Sakya lamas, showed an enormous amount of deference towards Deshin Shekpa. The Yongle Emperor came out of the palace in Nanjing to greet the Karmapa and did not require him to kowtow like a tributary vassal. According to Karma Thinley, the emperor gave the Karmapa the place of honor at his left, and on a higher throne than his own. Rossabi and others describe a similar arrangement made by Kublai Khan and the Sakya Phagpa lama, writing that Kublai would \"sit on a lower platform than the Tibetan cleric\" when receiving religious instructions from him.

Sino-Tibetan relations during the Ming dynasty\nThroughout the following month, the Yongle Emperor and his court showered the Karmapa with presents. At Linggu Temple in Nanjing, he presided over the religious ceremonies for the Yongle Emperor's deceased parents, while twenty-two days of his stay were marked by religious miracles that were recorded in five languages on a gigantic scroll that bore the Emperor's seal. During his stay in Nanjing, Deshin Shekpa was bestowed the title \"Great Treasure Prince of Dharma\" by the Yongle Emperor. Elliot Sperling asserts that the Yongle Emperor, in bestowing Deshin Shekpa with the title of \"King\" and praising his mystical abilities and miracles, was trying to build an alliance with the Karmapa as the Mongols had with the Sakya lamas, but Deshin Shekpa rejected the Yongle Emperor's offer. In fact, this was the same title that Kublai Khan had offered the Sakya Phagpa lama, but Deshin Shekpa persuaded the Yongle Emperor to grant the title to religious leaders of other Tibetan Buddhist sects.

Sino-Tibetan relations during the Ming dynasty\nIn his usurpation of the throne from the Jianwen Emperor (r. 1398–1402), the Yongle Emperor was aided by the Buddhist monk Yao Guangxiao, and like his father, the Hongwu Emperor, the Yongle Emperor was \"well-disposed towards Buddhism\", claims Rossabi. On March 10, 1403, the Yongle Emperor invited Deshin Shekpa, 5th Karmapa Lama (1384–1415), to his court, even though the fourth Karmapa had rejected the invitation of the Hongwu Emperor. A Tibetan translation in the 16th century preserves the letter of the Yongle Emperor, which the Association for Asian Studies notes is polite and complimentary towards the Karmapa. The letter of invitation reads.

Sino-Tibetan relations during the Ming dynasty\nIn order to seek out the Karmapa, the Yongle Emperor dispatched his eunuch Hou Xian and the Buddhist monk Zhi Guang (d. 1435) to Tibet. Traveling to Lhasa either through Qinghai or via the Silk Road to Khotan, Hou Xian and Zhi Guang did not return to Nanjing until 1407.

**Answer:** Nanjing was not the capital city of Yaxing Coach's headquarters.

Figure 6: **Good case 3 for QSGNN.**

in 1786"), and **iii)** geographic relationships ("Forest Acres borders Columbia"). The successful retrieval is due to the query-alignment and comprehensive understanding of contextual information. However, HippoRAG2 incorrectly retrieves Savannah county information due to over-reliance on "county" and "located on" seed nodes.

**Good Case 3**. Figure 6 presents a good case of QSGNN where it retrieves correctly and gives the correct answer but HippoRAG2 can't answer correctly because of the inaccurate retrieval. In this case, QSGNN successfully find the snippets for the "where the Yongle Emperor greeted the Karmapa", "the capitol city of Yaxing Coach's headquarters location" and the history information about "Yangzhou", which is the combination results of Multi-L KG and query-specific aggregation. HippoRAG2 retrieves redundant Sino-Tibetan relations documents. The failure occurs because HippoRAG2 prioritizes high PPR scores for entities "Yongle Emperor" and "Karmapa" without considering their contextual relationships.

## A.9 Bad Case Discussion and Limitations

We analyze three representative failure cases of QSGNN in document retrieval, each illustrating distinct limitations that inform our future research directions.

**Bad Case 1**. Figure 7 demonstrates QSGNN's failure to retrieve documents containing "Freikorps" due to a misspelled query terminology ("free crops" should be "Freikorps", we changed the "free crops" to "Freikorps" then QSGNN could retrieve correctly). This spelling error caused QSGNN to prioritize non-critical entities like "democratic government" and "Germany". We also check the pre-training data and fine-tuning data, there is no information about the associations between "free crops" and "Freikorps", which exacerbates the model's confusion.

**Query:** Who constituted the free crops in the location where the democratic government set up in Germany in 1919?
**Answer:** consisting largely of World War I veterans

**Gold docs:**
Freikorps\nIn the aftermath of World War I and during the German Revolution of 1918 -- 19, Freikorps consisting largely of World War I veterans were raised as right - wing paramilitary militias, ostensibly to fight on behalf of the government against the Soviet - backed German Communists attempting to overthrow the Weimar Republic. However, the Freikorps also despised the Republic and were involved in assassinations of its supporters. The Freikorps were widely seen as a precursor to Nazism, and many of their volunteers ended up joining the Nazi militia, the Sturmabteilung (SA). An entire series of Freikorps awards also existed.

Weimar Republic\nThe Weimar Republic is so called because the assembly that adopted its constitution met at Weimar, Germany from 6 February 1919 to 11 August 1919, but this name only became mainstream after 1933. Between 1919 and 1933 there was no single name for the new state that gained widespread acceptance, which is precisely why the old name ``Deutsches Reich ''continued in existence even though hardly anyone used it during the Weimar period. To the right of the spectrum the politically engaged rejected the new democratic model and cringed to see the honour of the traditional word`` Reich'' associated with it. The Catholic Centre party, Zentrum favoured the term ``Deutscher Volksstaat ''(`` German People's State'') while on the moderate left the Chancellor's SPD preferred ``Deutsche Republik ''(`` German Republic''). By 1925 ``Deutsche Republik ''was used by most Germans, but for the anti-democratic right the word`` Republik'' was, along with the relocation of the seat of power to Weimar, a painful reminder of a government structure that had been imposed by foreign statesmen, along with the expulsion of Kaiser Wilhelm in the wake of massive national humiliation. The first recorded mention of the term ``Republik von Weimar ''(`` Republic of Weimar'') came during a speech delivered by Adolf Hitler at a National Socialist German Worker's Party rally in Munich on 24 February 1929; it was a few weeks later that the term ``Weimar Republik ''was first used (again by Hitler) in a newspaper article. Only during the 1930s did the term become mainstream, both within and outside Germany.

**QSGNN**
**Retrieval:**
Strasbourg\nIn 1919, following the Treaty of Versailles, the city was restituted to France in accordance with U.S. President Woodrow Wilson's \"Fourteen Points\" without a referendum. The date of the assignment was retroactively established on Armistice Day. It is doubtful whether a referendum in Strasbourg would have ended in France's favour since the political parties striving for an autonomous Alsace or a connection to France accounted only for a small proportion of votes in the last Reichstag as well as in the local elections. The Alsatian autonomists who were pro French had won many votes in the more rural parts of the region and other towns since the annexation of the region by Germany in 1871. The movement started with the first election for the Reichstag; those elected were called "les députés protestataires\", and until the fall of Bismarck in 1890, they were the only deputies elected by the Alsatians to the German parliament demanding the return of those territories to France. At the last Reichstag election in Strasbourg and its periphery, the clear winners were the Social Democrats; the city was the administrative capital of the region, was inhabited by many Germans appointed by the central government in Berlin and its flourishing economy attracted many Germans. This could explain the difference between the rural vote and the one in Strasbourg. After the war, many Germans left Strasbourg and went back to Germany; some of them were denounced by the locals or expelled by the newly appointed authorities. The Saverne Affair was vivid in the memory among the Alsatians.

History of Germany during World War I\nDuring World War I, the German Empire was one of the Central Powers that lost the war. It began participation in the conflict after the declaration of war against Serbia by its ally, Austria - Hungary. German forces fought the Allies on both the eastern and western fronts, although German territory itself remained relatively safe from widespread invasion for most of the war, except for a brief period in 1914 when East Prussia was invaded. A tight blockade imposed by the Royal Navy caused severe food shortages in the cities, especially in the winter of 1916 -- 17, known as the Turnip Winter. At the end of the war, Germany's defeat and widespread popular discontent triggered the German Revolution of 1918 -- 19 which overthrew the monarchy and established the Weimar Republic.

East German uprising of 1953\nThe Uprising of 1953 in East Germany started with a strike by East Berlin construction workers on 16 June 1953. It turned into a widespread uprising against the German Democratic Republic government the next day. In Germany, the revolt is often called People's Uprising in East Germany (Volksaufstand in der DDR). It involved more than one million people in about 700 localities. 17 June was declared a day of national remembrance in West Germany up until reunification. Strikes and working class networks, particularly relating to the old Social Democratic Party of Germany, anti-fascist resistance networks and trade unions played a key role in the unfolding of the uprising.

History of Germany\nIn the early 1930s, the worldwide Great Depression hit Germany hard, as unemployment soared and people lost confidence in the government. In January 1933, Adolf Hitler was appointed Chancellor of Germany. His Nazi Party quickly established a totalitarian regime, and Nazi Germany made increasingly aggressive territorial demands, threatening war if they were not met. Remilitarization of the Rhineland came in 1936, then annexation of Austria in the Anschluss and parts of Czechoslovakia with the Munich Agreement in 1938, and further territory of Czechoslovakia in 1939.

History of Germany (1945–1990)\nThe intended governing body of Germany was called the Allied Control Council. The commanders - in - chief exercised supreme authority in their respective zones and acted in concert on questions affecting the whole country. Berlin, which lay in the Soviet (eastern) sector, was also divided into four sectors with the Western sectors later becoming West Berlin and the Soviet sector becoming East Berlin, capital of East Germany.

**Answer:** People's Uprising in East Germany

Figure 7: **Bad case 1 for QSGNN.**

**Query:** In which province is San Clemente, from the country where Fuser and Alberto meet the indigenous couple who were traveling to look for work?
**Answer:** Talca Province

**Gold docs:**
The Motorcycle Diaries (film)\nDuring their expedition, Guevara and Granado encounter the poverty of the indigenous peasants, and the movie assumes a greater seriousness once the men gain a better sense of the disparity between the ``haves ''(to which they belong) and the obviously exploited`` have - nots'' (who make up the majority of those they encounter) by traveling on foot. In Chile, for instance, they encounter a penniless and persecuted couple forced onto the road because of their communist beliefs. In a fire - lit scene, Guevara and Granado ashamedly admit to the couple that they are not out looking for work as well. The duo then accompanies the couple to the Chuquicamata copper mine, where Guevara becomes angry at the treatment of the workers.

San Clemente, Chile\nSan Clemente is a city and commune administered by the municipality of San Clemente, located in the Talca Province of Chile's Maule Region.

**QSGNN**
**Retrieval:**
C oquimbito\nCoquimbito is a rural district in the Maipú Department, Mendoza Province, Argentina. It is located in the southeast of the metropolitan area of Mendoza (the provincial capital), and is administratively part of the municipality of Maipú. The name refers to the Chilean port city of Coquimbo.

Colorado Territory\nA group of Cherokee crossed the South Platte and Cache la Poudre River valleys on their way to California in 1848 during the California Gold Rush. They reported finding trace amounts of gold in the South Platte and its tributaries as they passed along the mountains. In the south, in the San Luis Valley, early Mexican families established themselves in large land grants (later contested by the U.S.) from the Mexican government.

El Salvador\nEl Salvador lies in the isthmus of Central America between latitudes 13 ° and 15 ° N, and longitudes 87 ° and 91 ° W. It stretches 270 km (168 mi) from west - northwest to east - southeast and 142 km (88 mi) north to south, with a total area of 21,041 km (8,124 sq mi). As the smallest country in continental America, El Salvador is affectionately called Pulgarcito de America (the ``Tom Thumb of the Americas ''). The highest point in El Salvador is Cerro El Pital, at 2,730 metres (8,957 ft), on the border with Honduras.

Carpetania\nCarpetania was an ancient region of what is today Spain, located between the Sierra de Guadarrama, the mountains of Toledo, the river Guadiana and the mountain range of Alcaraz, including approximately, the present independent communities of Madrid and Castile-La Mancha. It was inhabited by the Carpetani, a pre-Roman tribe. To the south dwelt the Oretani, on the northeast were Celtiberians whose tribes are not further specified. On the northwest to the Vaccei and Vettones. This area was easily conquered by the Romans and quickly integrated culturally and politically. Thus it is practically unmentioned in the literature of the conquest. Its main urban nuclei (Toletum, corresponding to present Toledo; Complutum, the present Alcalá de Henares, Consabura, the present Consuegra, Segóbriga (Saelices, River basin) and Laminio) acquired municipal legal statutes soon after the Roman conquest.

Sebastian Cabot (explorer)\nSebastian Cabot (Italian and , ; , \"Gaboto\" or \"Cabot\"; 1474 – December 1557) was an Italian explorer, likely born in the Venetian Republic. He was the son of Italian explorer John Cabot (Giovanni Caboto) and his Venetian wife Mattea.

**Answer:** Mendoza Province, Argentina

Figure 8: **Bad case 2 for QSGNN.**

**Bad Case 2**. Figure 8 reveals QSGNN's limitation in processing specific term ("San Clemente", "Fuser", "Alberto"), which may not be well represented by QSGNN since they are absent from

**Query:** When was the person who Messi's goals in Copa del Rey compared to get signed by Barcelona?

**Answer:** June 1982

**Gold docs:**

FC Barcelona\nDespite being the favourites and starting strongly, Barcelona finished the 2006–07 season without trophies. A pre-season US tour was later blamed for a string of injuries to key players, including leading scorer Eto'o and rising star Lionel Messi. There was open feuding as Eto'o publicly criticized coach Frank Rijkaard and Ronaldinho. Ronaldinho also admitted that a lack of fitness affected his form. In La Liga, Barcelona were in first place for much of the season, but inconsistency in the New Year saw Real Madrid overtake them to become champions. Barcelona advanced to the semi-finals of the Copa del Rey, winning the first leg against Getafe 5–2, with a goal from Messi bringing comparison to Diego Maradona's goal of the century, but then lost the second leg 4–0. They took part in the 2006 FIFA Club World Cup, but were beaten by a late goal in the final against Brazilian side Internacional. In the Champions League, Barcelona were knocked out of the competition in the last 16 by eventual runners-up Liverpool on away goals.

FC Barcelona\nIn June 1982, Diego Maradona was signed for a world record fee of £5 million from Boca Juniors. In the following season, undercoach Luis, Barcelona won the Copa del Rey, beating Real Madrid. However, Maradona's time with Barcelona was short-lived and he soon left for Napoli. At the start of the 1984–85 season, Terry Venables was hired as manager and he won La Liga with noteworthy displays by German midfielder Bernd Schuster. The next season, he took the team to their second European Cup final, only to lose on penalties to Steaua București during a dramatic evening in Seville.

**QSGNN**
**Retrieval:**

FC Barcelona\nDespite being the favourites and starting strongly, Barcelona finished the 2006–07 season without trophies. A pre-season US tour was later blamed for a string of injuries to key players, including leading scorer Eto'o and rising star Lionel Messi. There was open feuding as Eto'o publicly criticized coach Frank Rijkaard and Ronaldinho. Ronaldinho also admitted that a lack of fitness affected his form. In La Liga, Barcelona were in first place for much of the season, but inconsistency in the New Year saw Real Madrid overtake them to become champions. Barcelona advanced to the semi-finals of the Copa del Rey, winning the first leg against Getafe 5–2, with a goal from Messi bringing comparison to Diego Maradona's goal of the century, but then lost the second leg 4–0. They took part in the 2006 FIFA Club World Cup, but were beaten by a late goal in the final against Brazilian side Internacional. In the Champions League, Barcelona were knocked out of the competition in the last 16 by eventual runners-up Liverpool on away goals.

FC Barcelona\nIt was announced in summer of 2012 that Tito Vilanova, assistant manager at FC Barcelona, would take over from Pep Guardiola as manager. Following his appointment, Barcelona went on an incredible run that saw them hold the top spot on the league table for the entire season, recording only two losses and amassing 100 points. Their top scorer once again was Lionel Messi, who scored 46 goals in the League, including two hat-tricks. On 11 May 2013 Barcelona were crowned as the Spanish football champions for the 22nd time, still with four games left to play. Ultimately Barcelona ended the season 15 points clear of rivals Real Madrid, despite losing 2–1 to them at the beginning of March. They reached the semifinal stage of both the Copa del Rey and the Champions League, going out to Real Madrid and Bayern Munich respectively. On 19 July, it was announced that Vilanova was resigning as Barcelona manager because his throat cancer had returned, and he would be receiving treatment for the second time after a three-month medical leave in December 2012.

Lionel Messi\nMessi opened the 2015 -- 16 season by scoring twice from free kicks in Barcelona's 5 -- 4 victory (after extra time) over Sevilla in the UEFA Super Cup. A subsequent 5 -- 1 aggregate defeat against Athletic Bilbao in the Supercopa de España ended their expressed hopes of a second sextuple, with Messi scoring his side's only goal. On 16 September, he became the youngest player to make 100 appearances in the UEFA Champions League in a 1 -- 1 away draw to Roma. On 26 September, Messi sustained an injury in Barcelona's match against Las Palmas; tests later confirmed that he suffered a tear in the medial collateral ligament of his left knee, ruling him out for six to eight weeks. He finally returned to the pitch on 21 November, making a substitute appearance in Barcelona's 4 -- 0 away win over rivals Real Madrid in El Clásico. Messi capped off the year by winning the 2015 FIFA Club World Cup Final on 20 December, collecting his fifth club trophy of 2015 as Barcelona went on to defeat River Plate 3 -- 0 in Yokohama. Messi also won the tournament's Silver Ball, despite missing the semi-final. On 30 December, Messi scored on his 500th appearance for Barcelona, in a 4 -- 0 home win over Real Betis.

Lionel Messi\nAfter a year at Barcelona's youth academy, La Masia, Messi was finally enrolled in the Royal Spanish Football Federation (RFEF) in February 2002. Now playing in all competitions, he befriended his teammates, among whom were Cesc Fàbregas and Gerard Piqué. After completing his growth hormone treatment aged 14, Messi became an integral part of the ``Baby Dream Team'', Barcelona's greatest - ever youth side. During his first full season (2002 -- 03), he was top scorer with 36 goals in 30 games for the Cadetes A, who won an unprecedented treble of the league and both the Spanish and Catalan cups. The Copa Catalunya final, a 4 -- 1 victory over Espanyol, became known in club lore as the partido de la máscara, the final of the mask. A week after suffering a broken cheekbone during a league match, Messi was allowed to start the game on the condition that he wear a plastic protector; soon hindered by the mask, he took it off and scored two goals in 10 minutes before his substitution. At the close of the season, he received an offer to join Arsenal, his first from a foreign club, but while Fàbregas and Piqué soon left for England, he chose to remain in Barcelona.

**Answer**: There is no connection between Messi's Copa del Rey goal (2007) and his signing by Barcelona

Figure 9: **Bad case 3 for QSGNN.**

both pre-training and fine-tuning corpora. The model deflect to general terms like "province" and "county", retrieving irrelevant documents.

**Bad Case 3**. Figure 9 shows a typical bad case for QSGNN. In this case the term like "Messi" and "Barcelona" dwarf the key evidence "Diego Maradona, who Messi can bring comparison to". The sequential dependency between finding "Diego Maradona" and subsequent evidence ("June 1982 transfer record") further complicates retrieval. It is difficult for query alignment to identify the correct information because huge amount of documents describing "Messi" and "Barcelona", which is also related to query, overwhelm the key snippets of "Diego Maradona".

**Limitation Discussion and Future Work**. Based on the analysis of the bad cases above, QSGNN is not a silver bullet for all multi-hop scenarios. It exhibits limitations when dealing with specific terminologies especially when they are not well presented in the corpora (see bad case 1, 2), and also struggles to deal the situation where the prevalent mis-related documents overwhelm the key evidence with sequential answer dependencies (see bad case 3). We think these limitations can be mitigated by two strategies: **i)** For the first issue, selecting seed nodes as domain-specific terminologies and sampling constrained subgraphs(depending on the layer of QSGNN, in our case the longest hop is 4), could prioritize relevant documents in retrieval, though this risks information loss. **ii)** For the second issue, integrating Chain-of-Thought(CoT) reasoning (Chen et al., 2019; Lyu et al., 2023) through query decomposition and iterative retrieval may mitigate the problem. However, the first solution is a tricky strategy and even may compromise the performance since subgraph sampling will lead to information loss. The second solution introduces CoT into framework, which is beyond the design topic of QSGNN. We will leave the combination of these two possible solutions without compromising the performance as the future work of QSGNN.

## A.10 Prompts for OpenIE

We present the prompts we used for OpenIE workflow. Figure 10 shows the prompts used for sentence extraction, Figure 11 shows the prompts for entity extraction, and Figure 12 shows the prompts for triple extraction. First, we extract sentences from each document. Then we identify

core entities from extracted sentences. At last, these entities are used to formulate triples from the document.

---

### Sentence Extraction Prompt

**Goal:**

Your task is to extract sentences from the given paragraph. Respond with a JSON list of sentences.

**Example:**

**- Input -:**
Radio City\nRadio City is India's first private FM radio station and was started on 3 July 2001.
It plays Hindi, English and regional songs. Radio City recently forayed into New Media in May 2008 with the launch of a music portal - PlanetRadiocity.com that offers music related news, videos, songs, and other music-related features.

**- Output -:**
{"sentences":
  [
    "Radio City is India's first private FM radio station and was started on 3 July 2001.",
    "It plays Hindi, English and regional songs.",
    "Radio City recently forayed into New Media in May 2008 with the launch of a music portal
     PlanetRadiocity.com that offers music related news, videos, songs, and other music-related features."
  ]
}

---

Figure 10: **Sentence extraction prompt for OpenIE.**

---

### Entity Extraction Prompt

**Goal:**

Your task is to extract named entities from the given paragraph. Respond with a JSON list of entities.

**Example:**

**- Input -:**
Radio City\nRadio City is India's first private FM radio station and was started on 3 July 2001.
It plays Hindi, English and regional songs. Radio City recently forayed into New Media in May 2008 with the launch of a music portal - PlanetRadiocity.com that offers music related news, videos, songs, and other music-related features.

**- Output -:**
{"named_entities":
  ["Radio City", "India", "3 July 2001", "Hindi", "English", "May 2008", "PlanetRadiocity.com"]
}

---

Figure 11: **Entity extraction prompt for OpenIE.**

### A.11 PROMPTS FOR SYNTHESIZED PRE-TRAINING DATA

The prompts used for generate pre-training data are shown in Figure 13 and Figure 14.

## Triple Extraction Prompt

**Goal:**

Your task is to construct an RDF (Resource Description Framework) graph from the given passages and named entity lists. Respond with a JSON list of triples, with each triple representing a relationship in the RDF graph.

Pay attention to the following requirements:
 - Each triple should contain at least one, but preferably two, of the named entities in the list for each passage.
 - Clearly resolve pronouns to their specific names to maintain clarity.

**Example:**

**- Input -:**
Convert the paragraph into a JSON dict, it has a named entity list.

Paragraph:
Radio City\nRadio City is India's first private FM radio station and was started on 3 July 2001.
It plays Hindi, English and regional songs. Radio City recently forayed into New Media in May 2008 with the launch of a music portal - PlanetRadiocity.com that offers music related news, videos, songs, and other music-related features.

Named_entities:
["Radio City", "India", "3 July 2001", "Hindi", "English", "May 2008", "PlanetRadiocity.com"]

**- Output -:**
{"triples":
  [
      ["Radio City", "located in", "India"],
      ["Radio City", "is", "private FM radio station"],
      ["Radio City", "started on", "3 July 2001"],
      ["Radio City", "plays songs in", "Hindi"],
      ["Radio City", "plays songs in", "English"],
      ["Radio City", "forayed into", "New Media"],
      ["Radio City", "launched", "PlanetRadiocity.com"],
      ["PlanetRadiocity.com", "launched in", "May 2008"],
      ["PlanetRadiocity.com", "is", "music portal"],
      ["PlanetRadiocity.com", "offers", "news"],
      ["PlanetRadiocity.com", "offers", "videos"],
      ["PlanetRadiocity.com", "offers", "songs"]
  ]
}

Figure 12: **Triple extraction prompt for OpenIE.**

## One-hop Question Generation Prompt

**Goal:**

Your task is to construct many question-answer pairs from the given passages with respond to a named entity lists. Respond with a JSON list of question-answer pairs, with each question asking one specific information about an entity and the answer is the ground truth of the question from the passage.

Pay attention to the following requirements:
- Each question should contain at least one, but preferably two, of the named entities in the list for each passage.
- Each answer should based on the fact proposed in the passage.
- The respond JSON list should have key as "query-answer pairs" and value is a list of dict with "question" and "answer" key.
- Clearly resolve pronouns to their specific names to maintain clarity.

**Example:**

**- Input -:**
Convert the paragraph into a JSON dict, it has a named entity list.

Paragraph:
Radio City\nRadio City is India's first private FM radio station and was started on 3 July 2001.
It plays Hindi, English and regional songs. Radio City recently forayed into New Media in May 2008 with the launch of a music portal - PlanetRadiocity.com that offers music related news, videos, songs, and other music-related features.

Named_entities: ["Radio City", "India", "3 July 2001", "Hindi", "English", "May 2008", "PlanetRadiocity.com"]

**- Output -:**
```
{"query-answer pairs": [
    {
      "question": "When was Radio City, India's first private FM radio station, started?",
      "answer": "Radio City was started on 3 July 2001."
    },
    {
      "question": "What features does PlanetRadiocity.com offer?",
      "answer": "PlanetRadiocity.com offers music related news, videos, songs, and other music-related  features."
    },
    {
      "question": "Is PlanetRadiocity.com associated with Radio City?",
      "answer": "Yes, PlanetRadiocity.com is associated with Radio City as it was launched by Radio City in May 2008."
    },
    {
      "question": "What is the significance of 3 July 2001 for Radio City?",
      "answer": "3 July 2001 is significant for Radio City as it marks the start of India's first private FM radio station."
    },
    {
      "question": "What was Radio City's expansion strategy in May 2008?",
      "answer": "Radio City's expansion strategy in May 2008 was the launch of a music portal, PlanetRadiocity.com."
    }
  ]
}
```

Figure 13: **One hop question generation prompt for OpenIE.**

---

**Two-hop Question Generation Prompt**

**Goal:**

Your task is to construct several 2-hop question-answer pairs from the given passages with respond to a entity lists. Respond with a JSON list of generated question-answer-doc triples, with each question is a 2-hop question asking indirectly about one specific information of an entity and the answer is either based on two passages or based on two descriptions of one specific passage.

I will provide:
- Two passages each with a entity list where the entity in the list can be found in the passage.
- A common entity list where each entity in the list can be found in both two passages.

Pay attention to the following requirements:
- Each question should ask about one entity in 2-hop way, meaning the answer can not be found directly but need to be found indirectly through the bridge of one other entity(common or uncommon).
- Each answer should based on the fact proposed in the passages and the supported passages should be given as number.
- The respond JSON list should have key as "question-answer-doc triples" and value contains: i) question: the generated 2-hop question, ii) answer: the answer to the question, iii) doc: the string contains all the the index number of the passage split by "," (note that index start with 0 and can only be 0 or 1!)
- Clearly resolve pronouns to their specific names to maintain clarity.

**Example:**

**- Input -:**
Convert the passages, entity lists with a common entity list into a JSON dict:

Passage one:
FC Barcelona\nBarcelona is the only European club to have played continental football every season since 1955, and one of three clubs to have never been relegated from La Liga, along with Athletic Bilbao and Real Madrid. In 2009, Barcelona became the first club in Spain to win the treble consisting of La Liga, Copa del Rey, and the Champions League. That same year, it also became the first football club ever to win six out of six competitions in a single year, thus completing the sextuple, comprising the aforementioned treble and the Spanish Super Cup, UEFA Super Cup and FIFA Club World Cup. In the 2014–15 season, Barcelona won another historic treble, making them the first club in European football to win the treble twice.

Entity list one: ['UEFA Super Cup', 'Champions League', 'Copa del Rey', 'FIFA Club World Cup', 'Spain', 'Barcelona', 'FC Barcelona', 'Spanish Super Cup', 'Athletic Bilbao', 'Real Madrid', 'La Liga']

Passage two:
FC Barcelona\nIt was announced in summer of 2012 that Tito Vilanova, assistant manager at FC Barcelona, would take over from Pep Guardiola as manager. Following his appointment, Barcelona went on an incredible run that saw them hold the top spot on the league table for the entire season, recording only two losses and amassing 100 points. Their top scorer once again was Lionel Messi, who scored 46 goals in the League, including two hat-tricks. On 11 May 2013 Barcelona were crowned as the Spanish football champions for the 22nd time, still with four games left to play. Ultimately Barcelona ended the season 15 points clear of rivals Real Madrid, despite losing 2–1 to them at the beginning of March. They reached the semifinal stage of both the Copa del Rey and the Champions League, goin g out to Real Madrid and Bayern Munich respectively. On 19 July, it was announced that Vilanova was resigning as Barcelona manager because his throat cancer had returned, and he would be receiving treatment for the second time after a three-month medical leave in December 2012.

Entity list two: ['December 2012', 'May 2013', 'Lionel Messi', 'Bayern Munich', 'Champions League', 'Copa del Rey', 'FC Barcelona', 'Pep Guardiola', 'Spanish', 'Tito Vilanova', 'Real Madrid', 'July']

Common entity list: ['Real Madrid', 'Champions League', 'Copa del Rey', 'FC Barcelona']

**- Output -:**
```
{"question-answer-doc triples": [
    {
    "question": "Which competition did FC Barcelona win in 2009 that made them the first club in Spain to achieve a specific historic feat?",
    "answer": "FC Barcelona won the treble consisting of La Liga, Copa del Rey, and the Champions League in 2009, becoming the first club in Spain to achieve this feat.",
    "doc": "0,1"
    },
    {
    "question": "Who was the manager of FC Barcelona when they won the Spanish football championship in May 2013?",
    "answer": "Tito Vilanova was the manager of FC Barcelona when they won the Spanish football championship in May 2013.",
    "doc": "1"
    },
    {
    "question": "What was the reason behind Tito Vilanova's resignation as FC Barcelona manager in July 2013?",
    "answer": "Tito Vilanova resigned as FC Barcelona manager because his throat cancer had returned and he needed to receive treatment for the second time.",
    "doc": "0,1"
    },
    {
    "question": "Which two clubs, along with FC Barcelona, have never been relegated from La Liga?",
    "answer": "Athletic Bilbao and Real Madrid are the two clubs, along with FC Barcelona, that have never been relegated from La Liga.",
    "doc": "0,1"
    },
  ]
}
```

Figure 14: **Two hop question generation prompt for OpenIE.**

