# OpenReview forum: "Query-Specific GNN: A Comprehensive Graph Representation Learning Method for Retrieval Augmented Generation"
_ICLR.cc/2026/Conference — ICLR 2026 Conference Withdrawn Submission_

### Official Review · Reviewer_1CN3 · 2025-10-16

**Soundness:** 1
**Presentation:** 2
**Contribution:** 2
**Rating:** 2
**Confidence:** 4

**Summary:**

The authors aim to improve performance on multi-hop questions in retrieval-augmented generation (RAG) settings. They propose a multi-level knowledge graph construction method that organizes documents into entity, chunk, and document levels. Subsequently, they introduce a graph neural network (GNN) to retrieve the most relevant documents from this multi-level KG while maintaining robustness against noise. The approach is evaluated on three multi-hop question answering datasets using nine baseline methods.

**Strengths:**

1. The performance improvement on high-hop questions is significant.

**Weaknesses:**

1. The paper contains multiple unjustified claims that require clarification, as outlined in the questions below. In summary, the paper does not provide adequate justification for why the core contributions, Multi-L KG and QSGNN, are effective.
2. The experimental section lacks comprehensive ablation studies to validate the individual contributions of each proposed component.
3. The writing needs improvement due to many undefined symbols and unclear equations.

**Questions:**

1. In line 195, the authors claim that "in most cases sentence is enough to act as a meaningful reference for questions." This statement lacks rigor and empirical justification.
2. In line 204, the authors did not mention the number of spans when linking adjacent chunks. Based on the text, the description suggests that the chunk graph is a line graph, but Figure 2 shows a different structure. Could the authors clarify this?
3. In line 208, the paragraph explains the advantages of the proposed Multi-L KG. However, the claims are presented intuitively without sufficient justification for their effectiveness. Could the authors explain the underlying mechanisms in more detail?
4. In Section 3.2, it is not clear to me why this design mitigates noise. Could the authors provide more explanation?
5. In line 227, the mathematical equations are difficult to understand due to many undefined symbols. For example, what are $\alpha$ and $\beta$? What is each equation trying to accomplish? What is the difference between $attn$ and $attn_{i, j}$?
6. What data are used by the baseline methods? Do they also use the same human-annotated data that was used to fine-tune the proposed QSGNN?
7. What is the performance before fine-tuning the proposed QSGNN with the human-annotated data?
There is no experiment demonstrating that Multi-L KG is a better type of KG. Is it possible to include more KG construction methods as baselines? Or could the authors show that baselines perform better when running on Multi-L KG compared to other types of KG?

---

### Official Review · Reviewer_bNAb · 2025-10-24

**Soundness:** 2
**Presentation:** 3
**Contribution:** 2
**Rating:** 4
**Confidence:** 4

**Summary:**

The paper proposes QS-GNN, a graph neural network (GNN) retriever for multi-hop question answering (QA). QS-GNN first generates a knowledge graph (KG) in the form of triples from a document corpus where links (relations) between extracted entities are augmented with document-level and chunk-level links (Multi-L KG). QS-GNN follows a two stage training on the graph: (1) pretraining based on synthetic data, (2) fine-tuning based on task-specific multi-hop QA. Experimental results on MuSiQue, 2Wiki, and HotpotQA show QS-GNN's effectiveness in multi-hop scenarios.

**Strengths:**

- S1) QS-GNN's approach on pretraining and fine-tuning on the Multi-L KG leverages the graph-based information generated from the corpus, capturing more fine-grained interactions between chunks.
-  S2) The designed GNN module operates both on intra- and inter-levels, enabling information flow across entities, chunks, and documents.

**Weaknesses:**

- W1) Query-specific GNNs are well-known in the KGQA literature (e.g., GNN-RAG and previous related works), limiting the novelty of the proposed solution.
-  W2) QS-GNN does not achieve significant improvements over HippoRAG2 (e.g., in Table 3). Additionally, Line 345 indicates that model selection was performed using test samples, suggesting potential overfitting to the test set.
- W3)  The pretraining stage (Section 3.3) uses (question, documents) pairs with document-level supervision, missing fine-grained entity-level information. In contrast, GFM-RAG pretrains on entity retrieval tasks. The authors should compare these different pretraining approaches to demonstrate the effectiveness of their document-level method.

**Questions:**

- Q1) Does QS-GNN process the full graph for each query, or do you extract subgraphs based on entity linking?

- Q2) Line 340 states QS-GNN uses a 2-layer GNN. How does a 2-layer architecture adequately handle multi-hop questions requiring >2 hops? Table 11 shows that additional layers actually degrade QS-GNN performance, could you explain this result?

---

### Official Review · Reviewer_jSfy · 2025-11-01

**Soundness:** 2
**Presentation:** 3
**Contribution:** 2
**Rating:** 4
**Confidence:** 3

**Summary:**

Existing RAG methods struggle with multi-hop QA tasks with complex semantic structures as well as noises introduced by retrieved knowledge information. This paper proposes a novel graph representation learning framework for multi-hop question retrieval, named Multi -L KG to model information in intra-inter levels for a more comprehensive understanding of complex multi-hop tasks. Besides, this paper designed QSGNN applying the mentioned KG to get query-specific knowledge to better understand the given question. QSGNN improves up to 33.8% on multi-hop scenarios, showing the effectiveness of the method especially on questions with more hops needed.

**Strengths:**

* This paper is well-written and experiments conducted are comprehensive.
* The method, QSGNN’s query-guided dual message passing mitigates noise sensitivity. It uses intra-level passing (capturing entity semantics and chunk coherence) and inter-level passing (fusing cross-granularity info), with both processes guided by query alignment. This prioritizes query-relevant nodes, reducing irrelevant noise.
* The experimental results show the great effectiveness of the proposed method.

**Weaknesses:**

* Multi-L KG integrates information from three levels, entity level, chunk level, document level. This method is very similar to the work[1], which builds GNN by structure-related and keyword-related nodes. More clarifications about this work or comparisons should be provided.

## References
[1] Li, Zijian, et al. "Graph neural network enhanced retrieval for question answering of large language models." Proceedings of the 2025 Conference of the Nations of the Americas Chapter of the Association for Computational Linguistics: Human Language Technologies (Volume 1: Long Papers). 2025.

**Questions:**

* Could the authors give detailed values of figure1?

---

### Official Review · Reviewer_Rxa5 · 2025-11-01

**Soundness:** 3
**Presentation:** 3
**Contribution:** 2
**Rating:** 4
**Confidence:** 4

**Summary:**

This paper introduces Query-Specific Graph Neural Network (QSGNN), a novel graph-based retrieval framework designed to enhance multi-hop retrieval in retrieval-augmented generation (RAG) systems. The authors first propose a Multi-information Level Knowledge Graph (Multi-L KG) that integrates multiple granularities of information (documents, entities, and chunks) to capture inter- and intra-document relations. On top of this structure, the proposed QSGNN applies query-guided message passing across both intra- and inter-level graph layers, allowing retrieval to focus on context relevant to the question while filtering out irrelevant noise. The model is pre-trained using synthesized data to improve generalization and robustness. Extensive experiments on multi-hop QA benchmarks (MuSiQue, 2Wiki, and HotpotQA) show that QSGNN consistently outperforms strong RAG baselines such as HippoRAG2, RAPTOR, and GraphRAG, especially in high-hop (3–4 hop) settings where retrieval difficulty increases sharply.

**Strengths:**

1 The paper targets a timely and technically challenging problem — multi-hop reasoning in retrieval-augmented LLMs
2 The query-specific message passing mechanism is both elegant and practical, as it dynamically aligns the retrieval process with the semantics of the user query.
3 The authors also show an awareness of efficiency trade-offs

**Weaknesses:**

1 The construction of the Multi-L KG is not deeply analyzed. Specifically, how the information levels are defined, linked, or weighted is described mostly procedurally rather than formally.
2 The experimental part only considers three datasets, which are very insufficient
3 The experimental part only involves several Graph RAG methods, e.g., GraphRAG and HippoRAG. However, there are a lot of more recent baselines that could be involved: UniKGQA, Think-on-Graph 2.0, RoG, R2-KG, G-retrieve, etc.
4. The method’s reliance on pre-built KGs and entity linking may limit applicability to open-domain settings where such structures are incomplete or unavailable.

**Questions:**

Could you elaborate on the generation process of the synthesized pre-training data and its contribution to final performance?
Have you tested QSGNN in open-domain RAG tasks where the answer is not in the KG?

---

### Note · Authors · 2025-11-26

I have read and agree with the venue's withdrawal policy on behalf of myself and my co-authors.